# MAP3Kinase-dependent SnRK2-kinase activation is required for abscisic acid signal transduction and rapid osmotic stress response

Yohei Takahashi [1], Jingbo Zhang[1,3], Po-Kai Hsu [1], Paulo H.O. Ceciliato[1], Li Zhang[1], Guillaume Dubeaux [1], Shintaro Munemasa[2], Chennan Ge[1], Yunde Zhao [1], Felix Hauser [1] & Julian I. Schroeder [1]*

Abiotic stresses, including drought and salinity, trigger a complex osmotic-stress and abscisic acid (ABA) signal transduction network. The core ABA signalling components are snf1-related protein kinase2s (SnRK2s), which are activated by ABA-triggered inhibition of type-2C protein-phosphatases (PP2Cs). SnRK2 kinases are also activated by a rapid, largely unknown, ABA-independent osmotic-stress signalling pathway. Here, through a combination of a redundancy-circumventing genetic screen and biochemical analyses, we have identified functionally-redundant MAPKK-kinases (M3Ks) that are necessary for activation of SnRK2 kinases. These M3Ks phosphorylate a specific SnRK2/OST1 site, which is indispensable for ABA-induced reactivation of PP2C-dephosphorylated SnRK2 kinases. ABA-triggered SnRK2 activation, transcription factor phosphorylation and SLAC1 activation require these M3Ks in vitro and in plants. M3K triple knock-out plants show reduced ABA sensitivity and strongly impaired rapid osmotic-stress-induced SnRK2 activation. These findings demonstrate that this M3K clade is required for ABA- and osmotic-stress-activation of SnRK2 kinases, enabling robust ABA and osmotic stress signal transduction.

[1] Division of Biological Sciences, Cell and Developmental Biology Section, University of California San Diego, La Jolla, CA 92093, USA. [2] Graduate School of Environmental and Life Science, Okayama University, Tsushima-Naka, Okayama 700–8530, Japan. [3] Present address: College of Resources and Environmental Sciences, National Academy of Agriculture Green Development, China Agricultural University, Beijing, PR China. *email: jischroeder@ucsd.edu

Limited water availability is one of the key factors that negatively impacts crop yields. The plant hormone abscisic acid (ABA) and the signal transduction network it activates, enhance plant drought tolerance through triggering of multiple cellular and developmental responses[1–3]. As plants are constantly exposed to changing water conditions, robustness of the ABA signal transduction cascade is important for plants to balance growth and drought stress resistance. Core ABA signalling components have been established[2–7]: ABA receptors PYR-ABACTIN RESISTANCE (PYR/PYL)/REGULATORY COMPONENT OF ABA RECEPTOR (RCAR) inhibit type-2C protein-phosphatases (PP2Cs)[4,5,8], resulting in activation of the snf1-related protein kinase2 (SnRK2) protein kinases SnRK2.2, 2.3 and OST1/SnRK2.6[4,5,9–11]. The SnRK2 kinases phosphorylate and thus regulate the activity of downstream components including transcription factors and ion channels[9,10,12–15], which leads to changes in gene expression and stomatal closure. Activation of SnRK2 protein kinases requires phosphorylation of the SnRK2 kinases themselves, and in vitro experiments using purified recombinant OST1/SnRK2.6 suggest that phosphorylation of the activation-loop is an important step[16]. However, it has remained unclear whether direct autophosphorylation and/or trans-phosphorylation by unknown protein kinases reactivate these SnRK2 protein kinases in response to stress.

Previous studies showed that ABA-dependent phosphorylation of substrate proteins by SnRK2s could be reconstituted using only recombinant PYR/RCAR ABA receptors, PP2Cs, and SnRK2 proteins[14,17,18]. Recombinant SnRK2 proteins used in these studies, unlike SnRK2s in plant cells, have high intrinsic kinase activities even before ABA treatment[16]. Moreover, ABA receptors, SnRK2 kinases, PP2Cs and targets have generally been added to reactions simultaneously[14,18]. Therefore it is not clear whether autophosphorylation accounts for the ABA-dependent SnRK2 reactivation after PP2C-dependent inhibition in planta.

The *Arabidopsis* genome encodes ten SnRK2 kinases, and at least nine of these are activated in response to osmotic stress[19]. Interestingly, rapid osmotic stress-induced activation of SnRK2 protein kinases can occur independently of ABA signalling[20]. The osmotic stress sensing mechanisms and upstream signal transduction mechanisms leading to SnRK2 activation remain to a large degree unknown in plants.

In the present study, a family of MAP kinase kinase kinases (M3Ks) is identified that is essential for reactivation of SnRK2 protein kinases after PP2C dephosphorylation. We show that the OST1/SnRK2.6 protein kinase cannot reactivate itself after dephosphorylation. Three independent reconstitution assays and in planta analyses show the function of these M3Ks in SnRK2 kinase reactivation and ABA signalling. Moreover interestingly, triple M3K knockout mutant analyses show that the identified M3Ks are required for the rapid osmotic stress activation of SnRK2 kinases, in a less-well understood, previously proposed, pathway parallel to ABA signalling.

## Results

**Isolation of ABA-insensitive MAPKK-kinase amiRNA mutants**. By unbiased forward genetic screening of seeds from over 1500 independent T2 artificial microRNA (amiRNA)-expressing lines in pools (~45,000 seeds screened) for ABA-insensitive seed germination, we isolated up to ~290 putative mutants. In secondary screening of the surviving putative mutants in the next (T3) generation, progeny from 25 of the putative mutant plants continued to show a clearly reduced ABA sensitivity, including seeds propagated from three *amiR-ax1117*-expressing plants (Fig. 1a–c). It is most likely that the three *amiRNA-ax1117*-expressing plants were the progeny of the same

amiRNA-expressing parent line. The *amiR-ax1117* is predicted to target five subgroup B Raf-like MAPKK-kinase (M3Ks) genes (Supplementary Fig. 1). Previously, in a redundancy-circumventing amiRNA pilot screen for impaired ABA inhibition of seed germination in *Arabidopsis*, we isolated putative mutants, including a M3K amiRNA-expressing line predicted to target seven MAPKK-kinases[21]. These seven putative target M3K genes overlap with four of the above *amiR-ax1117* target genes (Supplementary Fig. 1). Furthermore, in additional genetic screens for ABA-insensitive inhibition of seed germination using more than 2,000 pooled amiRNA-expressing lines (~50,000 seeds screened), we again isolated the previously isolated *m3k* amiRNA line two more times. The *amiR-ax1117* amiRNA and the *m3k* amiRNA target five and seven overlapping *Arabidopsis* Raf-like kinase members from subgroup B1 and B3 (Supplementary Fig. 1). Note that the *Arabidopsis* genome includes ~80 M3K genes and 22 B family M3K members[22]. Because SnRK2 protein kinase activation is a key step in ABA signalling, and based on prior findings described further below (Fig. 1f), we investigated ABA-activation of SnRK2 protein kinase activity in seedlings of the *m3k* amiRNA line by in-gel kinase assays. SnRK2 protein kinases are detected at apparent mobilities of 40–44 kDa in in-gel kinase assays[10,23]. Interestingly, ABA-activation of kinase activities was reduced by 60% in the *m3k* amiRNA line (Fig. 1d, e, Supplementary Fig. 2; $n = 3$ experiments).

**OST1/SnRK2.6 reactivation after dephosphorylation**. We investigated phosphorylation of purified recombinant GST-tagged OST1/SnRK2.6 protein kinase after dephosphorylation in vitro. To test whether OST1/SnRK2.6 could be re-activated by autophosphorylation, after dephosphorylation, the GST-OST1/SnRK2.6 protein bound on glutathione sepharose 4B resin was incubated with calf intestinal alkaline phosphatase (CIAP), and [γ-$^{32}$P]-ATP was added to the reaction after wash out of CIAP. Surprisingly, we found that OST1/SnRK2.6 showed very low autophosphorylation activity even after the protein phosphatase had been removed (Fig. 1f, lane 2; $n = 3$ experiments). Other ABA signalling protein kinases including the calcium-dependent protein kinases CPK6[24] and CPK23[25] and the MAP kinase MPK12[26,27] did not phosphorylate OST1/SnRK2.6 after dephosphorylation (Fig. 1f). Interestingly these results implied that autophosphorylation is not sufficient for OST1/SnRK2.6 reactivation following protein phosphatase exposure and removal. Therefore, another unknown protein kinase may be required for reversible ABA signal transduction.

We investigated whether the amiRNA-targeted M3Ks may directly activate OST1/SnRK2.6. In-gel kinase assays were carried out in vitro after incubation of the dephosphorylated His-OST1/SnRK2.6 with GST-tagged recombinant M3K kinase domains and His-tagged full-length M3Ks in the presence of ATP. Notably, three M3Ks from the subgroup B3, named M3Kδ1, δ6, and δ7, were found to strongly activate OST1/SnRK2.6, whereas the other M3Ks targeted by the corresponding amiRNA did not clearly activate OST1/SnRK2.6 under the imposed conditions in vitro (Fig. 1g and Supplementary Fig. 3; $n = 3$ experiments). OST1/SnRK2 kinase activation was not induced by an inactive mutant M3K kinase protein, M3Kδ6 (K775W) (Fig. 2a). Moreover, the M3Kδ1, δ6, and δ7 kinase domains directly phosphorylated the kinase inactive OST1/SnRK2.6 (D140A) mutant isoform (Fig. 2b and Supplementary Fig. 4). Note that a *Physcomitrella patens* protein kinase ARK showing similarity to these M3Ks was recently reported to phosphorylate a *Physcomitrella* SnRK2 kinase[28].

**M3Kδ1 phosphorylates a critical Ser171 for OST1activation**. Mass spectrometry analyses revealed that M3Kδ1 phosphorylated

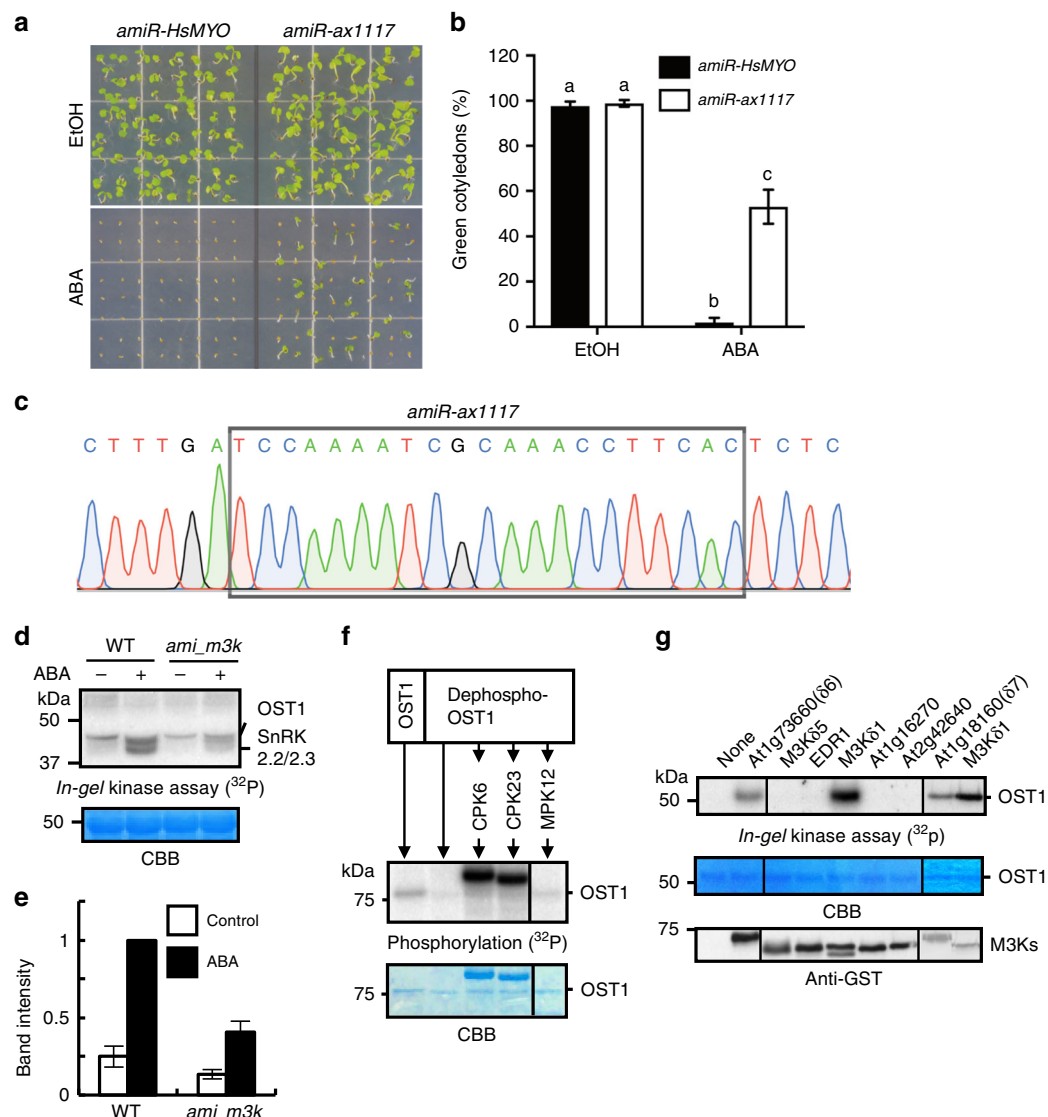

**Fig. 1 Identification of MAPKK-kinases that reactivate OST1/SnRK2 kinases by phosphorylation. a** Seeds of *amiR-HsMYO* wild-type (control line) or *amiR-ax1117* mutant were sowed on 1/2 MS medium containing 2 μM ABA, or 0.02% EtOH as control, for germination assays. Representative images showing seed germination after 6 days. **b** The percentage of seedlings showing green cotyledons was analyzed. Data represent mean ± s.d. $n = 4$ experiments. Each experiment included 64 seeds for each genotype. Letters at the top of columns are grouped based on two-way ANOVA and Tukey's test, $P < 0.05$. **c** Identification of the amiRNA sequence in *amiR-ax1117* plants. Black box labels the sequence of *amiR-ax1117*. The *amiR-ax1117* is predicted to include Raf-like protein kinase genes *M3Kδ5*, *M3Kδ7*, *M3Kδ1*, *M3Kδ6*, and *M3Kδ-CTR1* kinase (see Supplementary Fig. 1). **d** Wild-type (WT) and *m3k* amiRNA seedlings were incubated with 10 μM ABA for 15 min. *In-gel* kinase assays were performed using histone type III-S as a substrate. **e** SnRK2 band intensities as shown in **d** were measured using ImageJ, $n = 3$ experiments, error bars show ±s.e.m. **f**, Recombinant GST-OST1/SnRK2.6 protein was dephosphorylated by alkaline phosphatase in vitro and used for in vitro phosphorylation assays after incubation with CPK6, CPK23 or MPK12 protein kinases. Note visible autophosphorylation activity of CPK6 and CPK23. **g** Dephosphorylated recombinant His-OST1/SnRK2.6 protein was incubated with kinase domains of seven M3Ks and used for in-gel kinase assays (phylogenetic tree: see Supplementary Fig. 1). Note lanes on the left are from the same gel as lanes in the middle section.

the OST1/SnRK2.6 residues Ser171, Ser175, and Thr176 in the OST1-activation loop (Fig. 2c). We next focused on Ser171, because this site has not been found as an OST1/SnRK2.6 autophosphorylation site in vitro[16], consistent with our mass spectrometry analyses of OST1 (Fig. 2c, d). Using *Arabidopsis* mesophyll cell protoplasts as a transient expression system, consistent with a previous study[11], we found that substitution of this OST1/SnRK2.6 Ser171 by an alanine completely abrogated ABA-dependent activation of OST1/SnRK2.6 (Fig. 2e; $n = 3$ experiments).

Notably, the OST1-S171A mutation does not disrupt kinase activity in vitro, while another phosphorylation site mutation (S175A) disrupts kinase activity (Supplementary Fig. 5a, b). An OST1/SnRK2.6 T176A mutation does not disrupt kinase activity in vitro nor does the T176A mutation affect ABA activation of OST1/SnRK2.6 in vivo (Supplementary Fig. 5a, b). These results suggest that Ser171 plays an important role in ABA-activation of OST1/SnRK2.6 in plant cells. A potential phospho-mimic isoform of Ser171, OST1/SnRK2.6 (S171E) has no detectable kinase activity in mesophyll cells (Supplementary Fig. 5c). This is

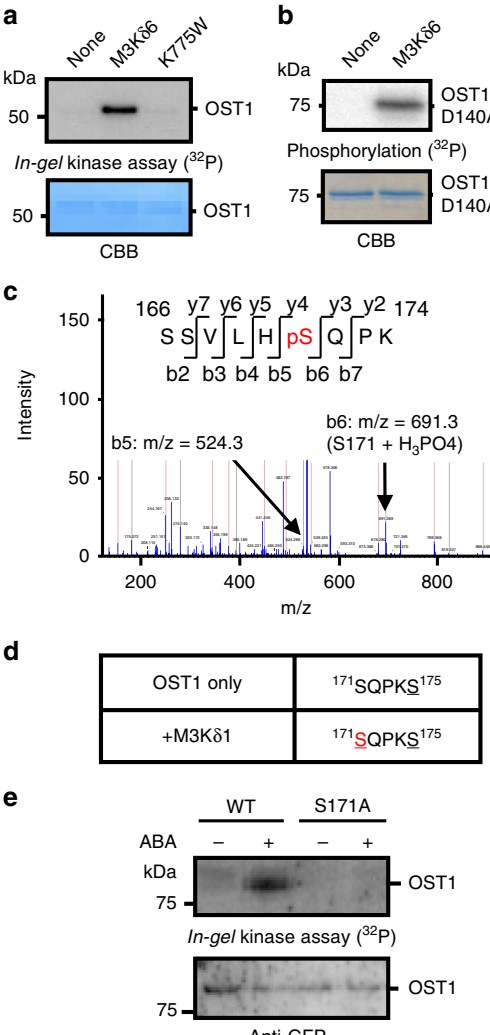

**Fig. 2 MAPKK-kinase-induced OST1/SnRK2.6 Ser171 phosphorylation is essential for ABA activation of OST1/SnRK2.6 activation. a** The inactive M3Kδ6 kinase domain mutant (K775W) did not reactivate His-OST1/SnRK2.6 in vitro. **b** Inactive GST-OST1/SnRK2.6-D140A kinase protein was incubated with M3Kδ6 kinase domain, and in vitro phosphorylation assays were performed with $^{32}$P-ATP. **c** Recombinant inactive OST1(D140A) and M3Kδ1 kinase domains were incubated with ATP. A mass spectrum of phosphorylated OST1 peptide (SSVLHpSQPK) is shown. pS indicates phosphorylated Ser171 of OST1(D140A). **d** Phosphorylation at Ser171 was not detectable after in vitro autophosphorylation of OST1/SnRK2.6, but was consistently phosphorylated in the presence of M3Kδ1. **e** OST1(S171A)-GFP was transiently expressed in *Arabidopsis* mesophyll cell protoplasts. Protoplasts were incubated with 10 μM ABA or control buffer for 15 min, and OST1/SnRK2.6 activities were analyzed by in-gel kinase assays.

consistent with a previously reported OST1/SnRK2.6 (S171D) mutant protein[11]. We further investigated the effect of ABA on phosphorylation of OST1-S171 in mesophyll cells. Ser171 is phosphorylated in plant mesophyll cells in response to ABA (Supplementary Fig. 6)[10,11].

We created transgenic *Arabidopsis* plants stably expressing OST1-HF (S171A) in the *ost1-3* background[29,30]. Expression of OST1-HF (S171A) did not rescue the ABA-insensitive stomatal conductance response and the low leaf temperature phenotype of the *ost1-3* mutant in two independent lines (Fig. 3a–c and Supplementary Fig. 7). Complementation of *ost1-3* with the wild-type OST1-HF isoform restored ABA-induced stomatal closing

and warm leaf temperatures (Fig. 3a–c and Supplementary Fig. 7), together indicating that Ser171 is required for OST1/SnRK2.6 function in stomatal closing (Fig. 3a–c and Supplementary Fig. 7).

Patch-clamp analyses of the *ost1-3* complementation lines showed the essential role of Ser171 for ABA-induced S-type anion channel activation in *Arabidopsis* guard cells (Fig. 3d, e). We further found that, in contrast to OST1-HF-expressing controls, OST1-HF (S171A) was not activated in *Arabidopsis* mesophyll cells in response to ABA in these stable homozygous transgenic plant lines (Fig. 3f).

**Reconstitution of early ABA signalling with MAPKK-kinases.** Previous studies have reconstituted ABA-dependent phosphorylation of OST1/SnRK2.6 substrates in vitro using recombinant proteins[14,18]. Recombinant OST1/SnRK2.6 has many phosphorylated sites and a significant protein kinase activity in vitro[16]. However, we find that prior dephosphorylated OST1/SnRK2.6 could unexpectedly not be re-activated by itself (Fig. 1f). We therefore hypothesized that these M3Ks have a role in reactivation of SnRK2 after inactivation by PP2C-mediated dephosphorylation. To test this, we pursued in vitro reconstitution experiments using recombinant proteins PYR1/RCAR11, the HAB1 PP2C, OST1/SnRK2.6 with or without M3Kδ6. In-gel kinase assays clearly showed that when HAB1-dependent OST1/SnRK2.6 dephosphorylation preceded ABA application, PYR1/RCAR11, HAB1, and OST1/SnRK2.6 could not recover OST1/SnRK2.6 activation (Fig. 4a; *n* > 3 experiments). Moreover, OST1/SnRK2.6 was no longer activated even after ABA treatment (Fig. 4a; *n* > 3 experiments). However, the OST1/SnRK2.6 kinase was clearly re-activated in response to ABA when M3Kδ6 was added to these reactions (Fig. 4b; *n* > 3 experiments). Consistent with these findings, in vitro reconstitution of ABA-dependent AKS1 transcription factor phosphorylation by OST1/SnRK2.6[18] was not observed when ABA was added after OST1/SnRK2.6 had been initially dephosphorylated by the PP2C HAB1 for 10 min (Fig. 4c, compare lanes 5, 6). Addition of M3Kδ6 restored ABA-induced His-AKS1 phosphorylation (Fig. 4c, compare lanes 7, 8).

**Reconstitution of ABA activation of SLAC1 requires M3Ks.** OST1/SnRK2.6-mediates activation of the S-type anion channel SLAC1 in *Xenopus* oocytes[12,13], and ABA-induced SLAC1 activation was reconstituted in oocytes[17]. These results strongly depended on artificial BiFC tags that force interaction of the SLAC1 channel with OST1/SnRK2.6 proteins[12,17], indicating that the BiFC tag might cause an unknown artificial effect. When expressing SLAC1 and OST1/SnRK2.6 proteins without any tag in oocyte experiments in the present study, SLAC1 was not significantly activated (Supplementary Fig. 8a–e). We found that SLAC1 was strongly activated when small amounts of *M3Kδ1*, *M3Kδ6*, or *M3Kδ7* cRNA were co-injected with OST1 into oocytes (Supplementary Fig. 8a–e; ratio of [M3K] to [*OST1*] cRNA = 1 to 10). However, the M3Ks did not activate SLAC1 in the absence of OST1/SnRK2.6 (Supplementary Fig. 8a–e), even when the injected M3K to *SLAC1* cRNA concentration ratio was 1 to 1. Furthermore, kinase inactive OST1/SnRK2.6 (D140A) does not activate SLAC1 in the presence of M3Kδ1 (Supplementary Fig. 8f, g).

In additional experiments, we co-injected cRNA for the ABA receptor *PYL9/RCAR1*, together with the *ABI1* PP2C, *OST1/SnRK2.6*, *SLAC1*, and M3Ks into oocytes, to test whether ABA-dependent SLAC1 anion channel activation could be reconstituted with these components. ABA could activate SLAC1 in oocytes only in the presence of low concentrations of either *M3Kδ1*, *M3Kδ6* or *M3Kδ7* mRNAs (Fig. 4d–f). Moreover, inactive M3K kinase mutant isoforms and inactive OST1

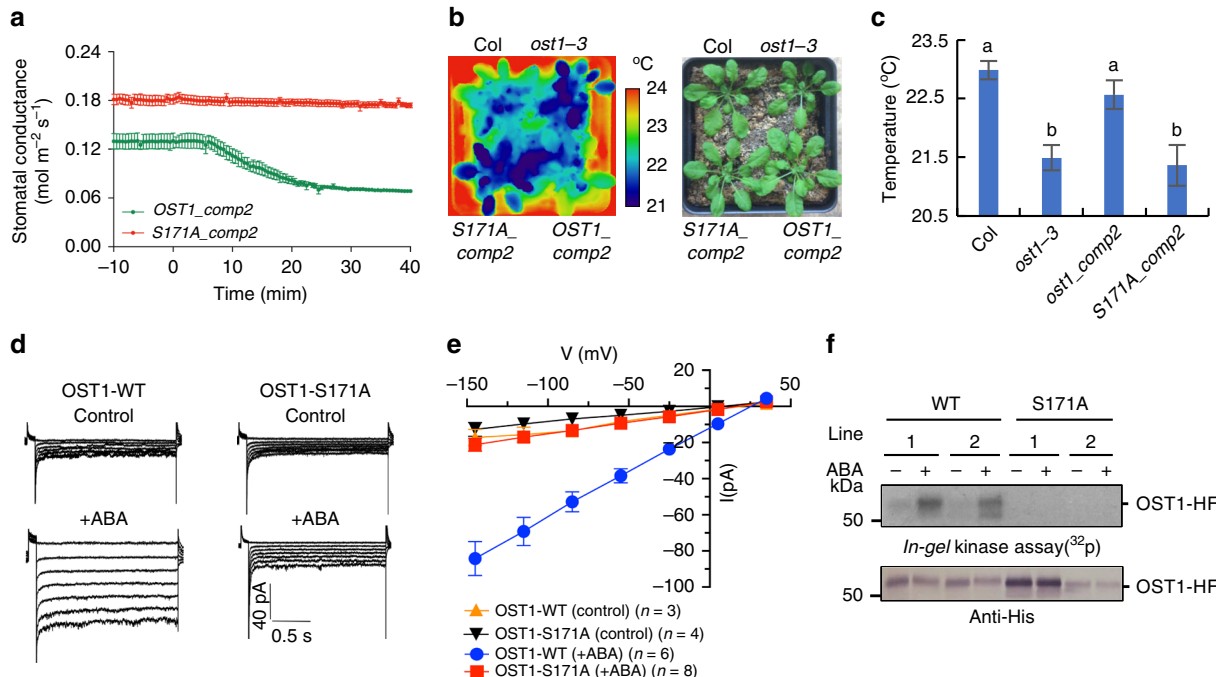

**Fig. 3 OST1/SnRK2.6 Ser171 is essential for ABA-induced stomatal closure and S-type anion channel activation *in planta*. a** Stomatal conductances were analyzed in intact detached leaves of stable transgenic *Arabidopsis* [*pUBQ10:OST1-HF/ost1-3* (*OST1-comp2*) and *pUBQ10:OST1-S171A-HF/ost1-3* (*S171A-comp2*)]. 2 μM ABA was applied to petioles at 0 min. Data presented are means ± s.e.m. (*n* = 4 leaves from four plants for each genotype). **b** Leaf temperatures of Col, *ost1-3*, *OST1-comp2* and *S171A-comp2* were measured by thermal imaging. Plants were sprayed with 20 μM ABA, and thermal images were taken after 3 h. The bright field image shows where leaves from neighboring plants overlapped. **c**, Leaf temperatures were measured by using Fiji software (*n* = 5 experiments, means ± s.e.m.). Letters at the top of columns are grouped based on one-way ANOVA and Tukey's test, *P* < 0.05. **d** ABA-activated S-type anion channel currents were investigated by patch-clamp analyses using guard cell protoplasts from the transgenic *Arabidopsis* lines *pUBQ10:OST1-HF/ost1-3* (*OST1-WT*) and *pUBQ10:OST1-S171A-HF/ost1-3* (*OST1-S171A*). **e** Average current-voltage relationship of S-type anion channel as shown in **d**. Data presented are means ± s.e.m. **f** Kinase activities of OST1(S171A) in mesophyll cells from stably-transformed homozygous transgenic plants were investigated by in-gel kinase assays. Protoplasts were incubated with 10 μM ABA for 15 min.

(S171A) disrupted reconstitution of SLAC1 activation (Supplementary Fig. 9). As SLAC1 plays an important role in ABA-induced stomatal closing, gas exchange experiments were pursued. *m3k* amiRNA plants show a reduced steady-state stomatal conductance and an ABA insensitivity in stomatal closure (Supplementary Fig. 10a, b).

The reduced steady-state stomatal conductance in the *m3k* amiRNA line indicates additional effects of this artificial microRNA and/or compensatory effects of impaired stomatal closing response mutants[31,32]. Higher order mutant combinations will be required to investigate this hypothesis. Based on the lower steady-state stomatal conductance, the impaired response to ABA (Supplementary Fig. 10a, b) and findings showing that ABA activation of S-type anion channels is an important mechanism for ABA-induced stomatal closing[24,33], we investigated ABA activation of S-type anion channels in guard cells. ABA (10 μM) caused typical ABA activation of S-type anion currents in guard cells of the wild-type (Col-0) and the *HsMYO* control line (Supplementary Fig. 10c–f). In contrast, ABA activation of S-type anion channels was impaired in guard cells of the *m3k* amiRNA line (Supplementary Fig. 10g, h). ABA signalling reconstitution (Fig. 4) and guard cell anion channel regulation analyses (Supplementary Fig. 10c–h) together suggest that the identified M3Ks provide a missing component of the early ABA signalling module.

**Higher order M3K mutants show ABA-insensitive phenotypes.** We isolated T-DNA insertion mutants [*m3kδ1* (SALK_048985), *m3kδ6-1* (SALK_004541), *m3kδ6-2* (SALK_001982), and *m3kδ7*

(SALK_082710)] (Fig. 5a). We also deleted large fragments of the *M3Kδ1* or *M3Kδ7* genes by CRISPR-Cas9 in the *m3kδ6-2* T-DNA knockout background (Fig. 5b), and a triple knockout mutant (*m3kδ1crispr m3kδ6-2 m3kδ7crispr*) was generated by crossing these lines (Fig. 5c) to analyze the physiological functions of these M3K genes. The *m3kδ1crispr m3kδ6-2 m3kδ7crispr* triple mutant showed a reduced ABA sensitivity phenotype in green cotyledon emergence from seeds (Fig. 5d, e). The double mutants *m3kδ1 m3kδ7* and *m3kδ6-2 m3kδ7* showed weaker ABA-insensitive phenotypes than the triple mutants (Supplementary Fig. 11a, b). Also, *m3kδ1/δ6-1/δ7* mutant seedlings showed a reduced ABA sensitivity in inhibition of primary root elongation on 1/2MS plates supplemented with ABA (Supplementary Fig. 11c).

We confirmed knockout of full-length expression of *M3Kδ1* and *M3Kδ7* in the T-DNA lines, while there was partial expression of the kinase domain of *M3Kδ6* in the *m3kδ6-1* line (Fig. 5f). Seed germination analyses showed reduced ABA sensitivity in the *m3kδ1 m3kδ6-1 m3kδ7* T-DNA insertion triple mutants (Fig. 5g, h). Another T-DNA allele for *M3Kδ6* for which the full length and kinase domain transcripts could not be amplified (Fig. 5a; *m3kδ6-2*) was considered. However, we could not isolate a viable *m3kδ1 m3kδ6-2 m3kδ7* triple mutant, possibly due to homozygous lethality, likely linked to an unknown second site mutation. Because the partial expression of the *M3Kδ6* kinase domain fragment was detected in the *m3kδ6-1* mutant (Fig. 5f), this kinase fragment may weaken the phenotypic effect.

To further test the function of these M3Ks, we created amiRNA lines predicted to target only the triple combination of *M3Kδ1*, *M3Kδ6*, and *M3Kδ7* and found that three independent amiRNA lines showed ABA-insensitivities in seed germination (Fig. 5i, j).

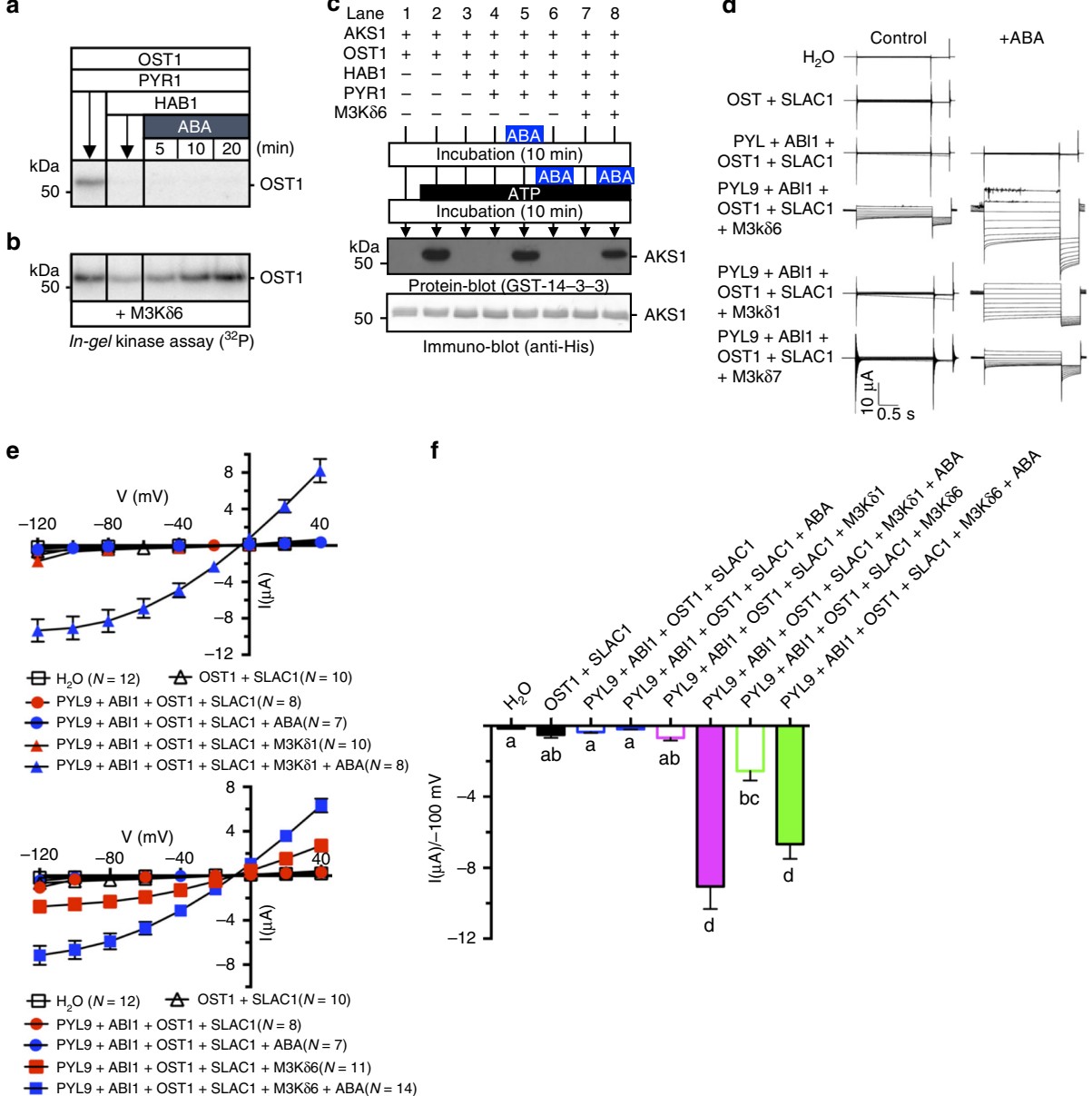

**Fig. 4 MAPKK-kinases are essential for ABA signalling module. a**, **b** In vitro reconstitution of ABA-induced OST1/SnRK2 activation without M3Kδ6 (**a**) or with M3Kδ6 (**b**). The recombinant proteins His-PYR1/RCAR11, His-OST1/SnRK2.6 without (**a**) or with (**b**) GST-M3Kδ6 kinase domain were mixed. After addition of His-HAB1, protein solutions were incubated for 10 min. Then, 50 μM ABA was added to the protein solution. Reactions were stopped at the indicated times. OST1/SnRK2.6 kinase activities were detected by in-gel kinase assays. **c** Recombinant His-PYR1/RCAR11, His-HAB1, His-OST1/SnRK2.6, His-AKS1, and GST-M3Kδ6 kinase domain were mixed as indicated above the gel. 50 μM ABA was added before (lane 5) or after (lanes 6 and 8) 10 min incubation at room temperature. Then, 100 μM ATP was added (lanes 2–8) to trigger phosphorylation reactions for 10 min. Note that M3Kδ6 is required for ABA-induced AKS1 phosphorylation when ABA is added 10 min after exposure to HAB1-PP2C-including mix (compare lanes 6 and 8). Reactions were stopped by addition of SDS-PAGE loading buffer. Phosphorylation of AKS1 is detected by binding of 14-3-3Phi (At1g35160) to the phosphorylated AKS1 protein[15]. AKS1 phosphorylation is shown by protein-blot (top), and protein amount is monitored by immuno-blot (bottom). **d**–**f** Reconstitution of ABA-activation of SLAC1 channels in *Xenopus* oocytes, in the presence or absence of M3Ks. **d** Representative whole-cell chloride current recordings of oocytes co-expressing the indicated proteins, without (control) or with injection of 50 μM ABA (+ABA). Currents were recorded in response to voltage pulses ranging from +40 mV to −120 mV in −20 mV steps with a holding potential at 0 mV and a final tail potential of −120 mV. **e** Mean current-voltage curves of oocytes co-expressing the indicated proteins, with or without injection of ABA. The symbols of H₂O control, OST1 + SLAC1, PYL9/RCAR1 + ABI1 + OST1 + SLAC1, PYL9/RCAR1 + ABI1 + OST1 + SLAC1 + ABA, and PYL9/RCAR1 + ABI1 + OST1 + SLAC1 + M3Ks overlapped. Single symbols are shown for some data points for better viewing. **f** Average SLAC1-mediated currents at −100 mV, co-expressing the indicated proteins, in the presence or absence of 50 μM ABA. Data from three independent batches of oocytes showed similar results. One representative batch of oocytes is shown, with the number of oocytes in that batch indicated in parentheses. H₂O, OST1 + SLAC1, PYL9/RCAR1 + ABI1 + OST1 + SLAC1, and PYL9/RCAR1 + ABI1 + OST1 + SLAC1 + ABA controls are the same data in both panels in **e** as the data are from the same oocyte batch. Error bars denote mean ± s.e.m. Means with letters (**a**, **b**, **c**, and **d**) are grouped based on one-way ANOVA and Tukey's multiple comparisons test, $P < 0.05$.

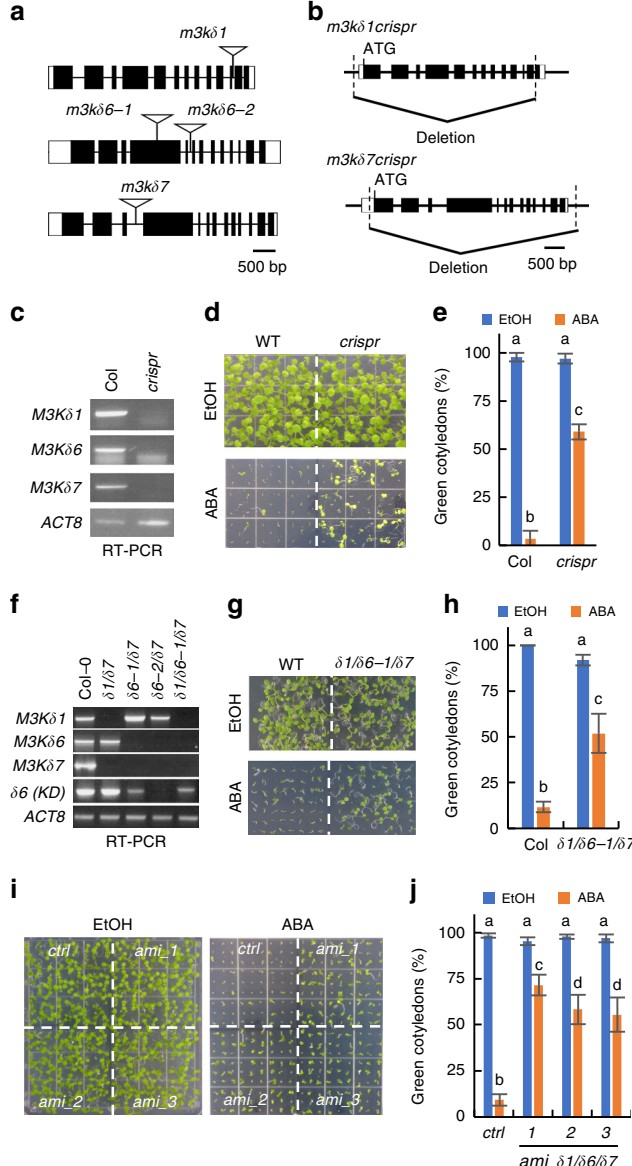

**Fig. 5 MAPKK-kinases are required for plant ABA response. a** Genome structures and T-DNA insertion sites of *M3K* genes are shown. **b** Genomic regions of CRISPR/Cas9-mediated *M3Kδ1* and *M3Kδ7* gene deletions are shown. These deletions were introduced in the *m3kδ6-2* T-DNA knockout mutant as a background. **c** RT-PCR assays show transcripts of kinase domains of *M3Ks* in the *m3kδ1crispr m3kδ6-2 m3kδ7crispr* triple mutant. **d** *m3kδ1crispr m3kδ6-2 m3kδ7crispr* triple mutant seedlings were grown on 1/2 MS plates supplemented with 2 μM ABA or ethanol (control) for 16 days. **e** Seedlings showing green cotyledons as in **d** were counted. *n* = 3 (EtOH) and *n* = 4 (ABA) experiments, means ± s.d., 45 seeds per genotype were used in each experiment. **f** RT-PCR shows *M3Kδ1*, *δ6*, and *δ7* expression in the indicated *m3k* T-DNA insertion mutants. *δ6(KD)* refers to primers that amplify the *M3Kδ6* kinase domain in the *m3kδ6-1* T-DNA line. **g** *m3kδ1 m3kδ6-1 m3kδ7* T-DNA triple mutant plants were grown on 1/2MS plates supplemented with 0.8 μM ABA for 9 days. **h** Seedlings showing green cotyledons as in **g** were counted. *n* = 3, means ± s.d., 60-88 seeds were used per genotype in each assay. **i** Three amiRNA lines targeting M3Kδ1, δ6, and δ7 were grown on 1/2MS plates supplemented with EtOH (control) or 2 μM ABA for 9 days. As a control line, the *amiRNA-HsMYO* line[21] was used. **j** Seedlings showing green cotyledons as in **i** were counted. *n* = 3 (EtOH) and 4 (ABA) experiments, means ± s.d., 81 seeds per genotype were analyzed in each experiment. (**e**, **h**, and **i**) Letters at the top of columns are grouped based on two-way ANOVA and Tukey's test, *P* < 0.05.

Together these results support that these M3Ks have a function in ABA responses.

**ABA- and osmotic stress- SnRK2 activations require M3Ks.** In-gel kinase assays showed that ABA-induced activation of SnRK2 kinase in the *m3kδ1crispr m3kδ6-2 m3kδ7crispr* triple was slightly less strong than in wild-type plants (Fig. 6a, b; *n* = 4 experiments). We further found a slightly reduced ABA activation of SnRK2 kinase activity in the T-DNA insertion *m3kδ1 m3kδ6-1 m3kδ7* triple mutant compared to wild-type controls (Fig. 6c, d, Supplementary Fig. 12; *n* = 4 experiments), similar to the *m3kδ1crispr m3kδ6-2 m3kδ7crispr* triple knockout mutant allele findings. Osmotic stress is known to rapidly activate OST1/SnRK2.6 independent of ABA signalling[20]. Interestingly, we found that 15 min osmotic stress-induced SnRK2 activation was strongly impaired in these two independent *m3k* triple mutant alleles, and this impairment was stronger than that in response to ABA application (Fig. 6a–d, Supplementary Fig. 12; *n* = 4 experiments per allele).

In-gel kinase assays suggest that the M3Ks have a major role in osmotic stress signalling in *Arabidopsis* (Fig. 6a–d). We therefore investigated osmotic-stress responses of the *m3k* double and triple mutants and the *m3k* amiRNA lines, and found that they showed reduced sensitivity to osmotic-stress in seed germination assays (Supplementary Fig. 13). SnRK2 gene functions are highly redundant in mediating osmotic stress resistance[34]. At least nine members out of the ten *Arabidopsis* SnRK2 proteins are activated by osmotic stress through unknown mechanisms, while three members (SnRK2.2/2.3/2.6) are major ABA-activated SnRK2s[19,23]. In vitro in-gel kinase assays showed that M3Kδ1 strongly activated SnRK2.2 and 2.3 (Fig. 6e) as well as OST1/SnRK2.6 (Fig. 1g). SnRK2.3 was also activated by M3Kδ6 and M3Kδ7 (Supplementary Fig. 14a). We also found that SnRK2.2 (S180A) and SnRK2.3 (S172A), which have a mutation corresponding to OST1/SnRK2.6 (S171A), are not activated by ABA in mesophyll cell protoplasts in contrast to WT SnRK2.2 and WT SnRK2.3 (Supplementary Fig. 14b). M3Kδ1 also activated SnRK2.4 kinase in vitro that is known to be activated by osmotic stress[19].

Co-immunoprecipitation of M3Kδ6- and OST1/SnRK2.6-expressed in mesophyll cell protoplasts did not show a clear interaction (Supplementary Fig. 15a). Protein kinase interactions are often transient and do not show co-immunoprecipitation with their targets[35]. BiFC analyses can detect transient interactions in plant cells. Quantitative BiFC experiments provide evidence that M3Kδ6 and M3Kδ7 bind to OST1/SnRK2.6, SnRK2.2, SnRK2.4, and SnRK2.10 in plant cells with different efficiencies (Supplementary Fig. 15b–e). We further observed that the M3Kδ6-FLAG protein band in SDS-PAGE gels was slightly shifted in response to 15 min osmotic stress treatment in mesophyll cell protoplasts, suggesting an osmotic stress-dependent post-translational modification of M3Kδ6 (Fig. 6f; *n* = 3).

## Discussion

In the present study, a combination of genetic screening for functional redundancy in abscisic acid responsiveness and multiple biochemical and signal transduction analyses in vitro and in planta have identified and characterized members of the Raf-like MAPKK-kinase δ B3 family that are required for full activation of SnRK2 protein kinases in abscisic acid signal transduction in vitro (Fig. 1g and Fig. 4a–c), in a reconstitution system (Fig. 4d–f) and in planta (Figs. 1a–e, 5d–j and 6a–d). Triple mutants in the *M3Ks* *AtM3Kδ1*, *M3Kδ6*, and *M3Kδ7* show impaired ABA- and osmotic stress-responses. As the *Arabidopsis* genome includes 80 MAPKK-kinases[22], of which 22 MAPKK-kinases are in the B

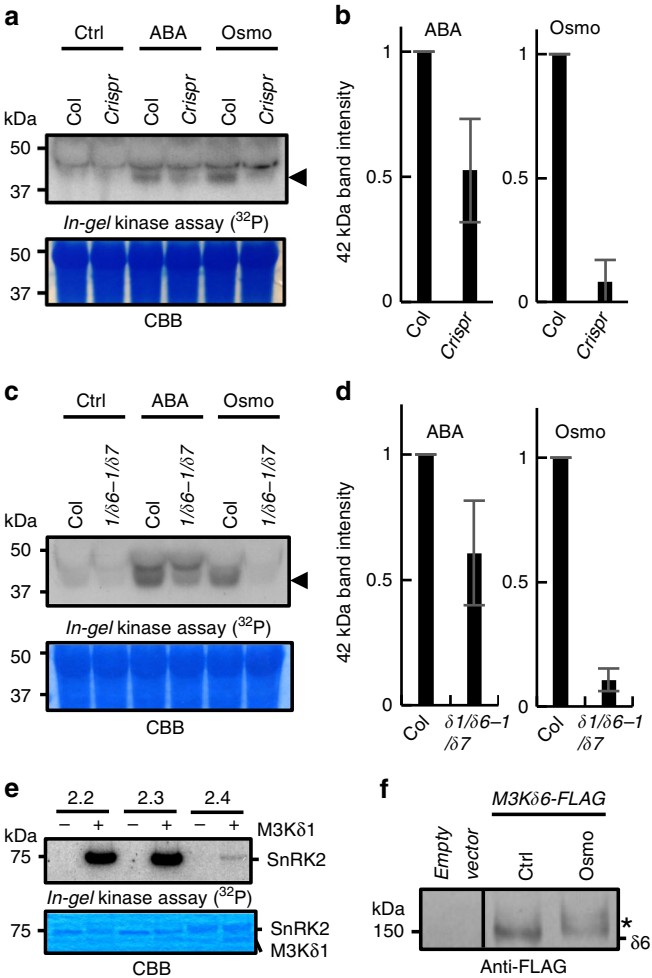

**Fig. 6 MAPKK-kinases mediate ABA- and osmotic stress-induced SnRK2 activation *in planta*.** **a** *m3kδ1crispr m3kδ6-2 m3kδ7crispr* triple mutant seedlings were incubated in 10 μM ABA or 0.3 M mannitol (Osmo) for 15 min. SnRK2 activities were tested by in-gel kinase assays. Arrowhead shows SnRK2 activity[23]. **b** Normalized band intensities as shown in **a** were measured by using ImageJ. $n = 4$, means ± s.e.m. **c** *m3kδ1 m3kδ6-1 m3kδ7* T-DNA triple mutant seedlings were incubated in 10 μM ABA or 0.3 M mannitol (Osmo) for 15 min. SnRK2 activities were analyzed by *in-gel* kinase assays. **d** Normalized band intensities as shown in **c** were measured by using ImageJ. $n = 4$ experiments, means ± s.e.m. **e**, Recombinant GST-tagged *Arabidopsis* SnRK2 protein kinases were incubated with M3Kδ1 kinase domain. SnRK2 kinase activities were analyzed by *in-gel* kinase assays. **f** *M3Kδ6-FLAG* was transiently expressed in *Arabidopsis* mesophyll cell protoplasts. Protoplasts were incubated in 0.8 M mannitol (Osmo) for 15 min. M3Kδ6 proteins were detected by immuno-blot using anti-FLAG antibody. In the Osmo lane, the M3Kδ6 band showed a slight mobility shift as indicated by an asterisk.

subgroup, it is conceivable that additional members of this family contribute to ABA responses and that higher order mutants will cause enhanced ABA insensitivity. Previous studies suggest that other MAPKK-kinases, than those identified here, are involved in aspects of ABA signalling through a MAP3K-MAP2K-MAPK cascade[36–38] or through unknown pathways[39,40].

Dephosphorylation of the OST1/SnRK2.6 kinase was unexpectedly found not to result in OST1/SnRK2.6 reactivation by SnRK2 autophosphorylation alone. The identified M3Kδs, but not other analyzed CPK and MPK12 protein kinases that function in ABA signalling[24–27], were found to be required for reactivation of OST1/SnRK2.6. Moreover, the M3Kδ1 kinase greatly enhances the

activities of other ABA signalling protein kinases SnRK2.2 and SnRK2.3 (Fig. 6e). Furthermore, M3Kδs reactivate OST1/SnRK2.6 through phosphorylation of Ser171 in OST1/SnRK2.6. The Ser171 residue in OST1/SnRK2.6 is essential for ABA responses in planta (Fig. 3 and Supplementary Fig. 7), but OST1/SnRK2.6 cannot autophosphorylate this Ser-171 residue (Fig. 2)[11,16]. These data point to the model that the M3Kδs identified here are essential for SnRK2 kinase reactivation and thus robust ABA responses in plants. Higher order M3K mutants and further experiments will be needed to investigate M3K-dependent Ser171 phosphorylation of OST1/SnRK2.6 in planta.

A previous proof-of-concept screen using artificial microRNAs that target multiple homologous genes isolated a plant predicted to target seven M3Ks of the B-family[21]. A *Physcomitrella* single gene encoding a M3K, ARK, was also identified which functions in SnRK2 activation[28]. Recent studies show that ARK kinase is required for *Physcomitrella* ABA and drought stress responses including phosphorylation of transcription factors through SnRK2 kinases[41,42]. Here, in forward genetic screening we have isolated amiRNA-expressing lines that target M3K members of the B family (Fig. 1a–c and Supplementary Fig. 1). In the present study, we show that for prior dephosphorylated SnRK2 kinases, we could robustly reconstitute ABA-activation of OST1/SnRK2.6 and the SLAC1 anion channel only in the presence of M3Kδs in vitro and in *Xenopus* oocytes (Fig. 4). The present experiments reveal that autophosphorylation cannot alone reactivate the SnRK2 kinases. These data suggest that these M3Ks are a missing component of the early ABA signalling module in plants.

Osmotic stress is known to rapidly activate SnRK2 protein kinases[20,34,43]. Rapid osmotic stress signalling includes a prominent ABA-independent pathway that leads to activation of transcription factors[44,45]. However, the upstream osmotic stress signalling mechanisms remain incompletely understood. Recent studies suggest that PP2Cs involved in ABA signalling dephosphorylate SnRK2.4[46–48]. The M3K ARK is required for osmotic stress tolerance in *Physcomitrella*[28,42]. Interestingly, the identified M3Kδs play a critical role in the rapid osmotic stress activation of SnRK2 protein kinases (Fig. 6a–d). In *m3k* triple mutants, 15 min short term osmotic stress activation of SnRK2 is greatly impaired in planta. This impairment in rapid osmotic stress activation of SnRK2 protein kinases is prominent in the investigated *m3kδ1/δ6/δ7* triple mutant alleles, in contrast to that of ABA activation of SnRK2 kinases (Fig. 6a–d). Further research will be needed to determine whether higher order *m3k* mutants further impair the ABA response. To start testing this hypothesis, we created *m3kδ1/δ5/δ6-1/δ7* quadruple mutant plants and found that they show a stronger ABA-insensitive phenotype in seed germination than the triple mutant (Supplementary Fig. 16a). Triple mutant plants, which include the weak allele *m3kδ6-1 (m3kδ1/δ6-1/δ7)*, did not show a clear phenotype in ABA-induced stomatal closing using a robust method of gas exchange analyses. The public eFP Browser shows a prominent guard cell expression of *M3Kδ5* (Supplementary Fig. 16b). *M3Kδ5* is targeted by the *m3k* amiRNA (Supplementary Fig. 1), which shows an ABA-insensitive stomatal closing (Supplementary Fig. 10a, b) and impairs ABA activation of S-type anion channels (Supplementary Fig. 10c–h). Higher order mutants will be required to further investigate M3K functions in ABA-induced stomatal closing. The requirement of M3Kδs for the rapid osmotic stress response suggests that these M3Kδs also mediate osmotic stress signal transduction before the slower onset of ABA concentration increase 4–6 h after exposure to osmotic stress[49]. These findings are consistent with previous observations of an ABA-independent osmotic stress-triggered SnRK2 signal transduction pathway[20,43,50]. The present study points to a model in which the identified M3Kδ protein kinases

may act as a convergence point of rapid osmotic stress signalling and prolonged abscisic acid signal transduction.

Osmotic and salt stresses induce a rapid cytosolic $Ca^{2+}$ increase[51–54]. An ABA-independent osmotic stress signalling pathway has been characterized that triggers rapid gene expression[44,55]. Recent research shows that the *Arabidopsis* NGATHA1 transcription factor mediates the ensuing drought stress-induced ABA accumulation through enhanced expression of the ABA biosynthesis *NINE-CIS-EPOXYCAROTENOID DIOXYGENASE, NCED3*[56]. Gel shift assays indicate that osmotic stress causes a rapid post-translational modification of M3Kδ6 (Fig. 6f). Our results reveal a key component by which plants respond initially to osmotic stress before measurable stress-induced ABA concentration increases in roots. Furthermore, interestingly, *m3k* amiRNA lines impair robust ABA activation of SnRK2 kinases in planta. Further research will be required to elucidate the presently unknown mechanisms between osmotic stress sensing and M3Kδ-dependent activation of SnRK2 protein kinases.

## Methods

**Genetic screening for ABA response mutants.** Using amiRNA libraries[21], we screened amiRNA lines for ABA-insensitive seed germination phenotypes using 1/ 2 MS plate supplemented with 2 μM ABA[57]. The underlying amiRNA sequences were identified from genomic DNA by PCR and sequencing (*m3k* amiRNA: 5′- TTGGAGCCATCCATTCAGCCG-3′, *amiR-ax1117*: 5′- TCCAAAATCG-CAAACCTTCAC-3′). We used an amiRNA line targeting human *myosin 2* gene (*HsMYO2*) as a control.

**In vitro dephosphorylation and phosphorylation assays.** Ten microgram of GST-OST1/SnRK2.6 proteins were bound to glutathione sepharose 4B beads and incubated with 30 U CIAP for 2 h at room temperature. The beads were washed with T-TBS (50 mM Tris-HCl pH 7.5, 150 mM NaCl, 0.05% Tween-20) three times, and GST-OST1 protein was eluted in 30 μL elution buffer (50 mM Tris-HCl pH 8.0, 10 mM reduced glutathione). Five microliter of GST-OST1 solution was added in phosphorylation buffer [50 mM Tris-HCl pH 7.5, 10 mM MgCl$_2$, 2 μM free $Ca^{2+}$ buffered by 1 mM EGTA and CaCl$_2$ (https://somapp.ucdmc.ucdavis.edu/pharmacology/bers/maxchelator/CaMgATPEGTA-NIST.htm), 0.1% Triton X-100, and 1 mM DTT] with or without 1 μg of the protein kinases CPK6, CPK23, MPK12 or 0.1 μg of the indicated MAPKK kinases (M3Ks). The phosphorylation reactions were started by addition of 200 μM ATP and 1 μCi [γ-$^{32}$P] ATP. After 60 min incubation at room temperature, these reactions were stopped by addition of SDS-PAGE loading buffer. Note that the mobilities of recombinant and transgenic proteins in the present study depend on the linked tags. For example, the OST1/SnRK2.6 6xHis-tag also includes sequences including thrombin and enterokinase cleavage sites and restriction enzyme sites in the pET-30a(+) vector used for *E.coli* expression of OST1/SnRK2.6 in Figs. 1g, 4a and b. Primer sequences used for cloning in this study are provided in Supplementary Table 1.

**In-gel kinase assays.** Fifteen to twenty *Arabidopsis* seedlings (7–9-day-old) grown on 1/2 MS plates were treated with 10 μM ABA or 0.3 M mannitol for 15 min at room temperature and grinded with a pestle and mortar in 400 μL extraction buffer (50 mM MOPS-KOH pH 7.5, 100 mM NaCl, 2.5 mM EDTA, 10 mM NaF, 2 mM dithiothreitol, 1 mM phenylmethylsulfonyl fluoride, 10 μM leupeptin) on ice. After 10 min centrifugation at 13,000 × g, the supernatants were transferred to new tubes, and proteins were precipitated by acetone precipitation. Proteins were dissolved in SDS-PAGE loading buffer and separated in 9% acrylamide gels. In-gel kinase assays were performed as described previously[58]. In brief, gels were incubated in washing buffer (25 mM Tris-HCl pH 8.0, 0.5 mM DTT, 0.1 mM Na$_3$VO$_4$, 5 mM NaF, 0.5 mg ml$^{-1}$ BSA, and 0.1% Triton X-100) for 30 min three times and in renaturation buffer (25 mM Tris-HCl pH 8.0, 1 mM DTT, 0.1 mM Na$_3$VO$_4$, and 5 mM NaF) for 30 min once. Gels were further incubated in renaturation buffer at 4 °C overnight followed by further incubation in reaction buffer (50 mM Tris-HCl pH 7.5, 10 mM MgCl$_2$, 2 mM DTT, and 1 mM EGTA) for 30 min. Phosphorylation reactions were carried out in reaction buffer with 50 μCi [γ-$^{32}$P]-ATP for 60 min at room temperature. Gels were washed in 5% trichloroacetic acid and 1% phosphoric acid four times for 30 min each. Storage phosphor screens or X-ray films were used for detection.

**In vitro reconstitution of ABA signalling.** 0.43 μmol His-OST1/SnRK2.6, 0.17 μmol His-PYR1/RCAR11 and 0.06 μmol GST-M3Kδ6 kinase domain were incubated in 200 μL phosphorylation buffer (50 mM Tris-HCl pH 7.5, 10 mM MgCl$_2$, 0.1% Triton X-100, and 1 mM DTT) with 200 μM ATP for 10 min, and 20 μL solution was transferred to a new tube and 10 μL 3xSDS-PAGE loading buffer was added to stop the reaction. Then, 0.01 μmol His-HAB1 was added to the reaction solution, and 20 μL solution were transferred to a new tube to stop the reaction by

addition of 10 μL 3xSDS-PAGE loading buffer after 10 min incubation. 50 μM ABA was added to the reaction and 20 μL reactions were transferred to new tubes to stop the reaction after 5, 10, or 30 min incubation. Proteins were separated by SDS-PAGE, and OST1/SnRK2.6 activity was detected by in-gel kinase assays.

**Identification of OST1/SnRK2.6 phosphorylation sites.** Thirty microgram GST-OST1/SnRK2.6(D140A) and 2.5 μg GST-M3Kδ1 kinase domain were incubated in phosphorylation buffer (50 mM Tris-HCl pH 7.5, 10 mM MgCl$_2$, 0.1% Triton X-100, and 1 mM DTT) with 1 mM ATP for 2 h at room temperature. Proteins were precipitated by acetone precipitation and dissolved in SDS-PAGE loading buffer. After SDS-PAGE and CBB staining, protein bands of GST-OST1/SnRK2.6(D140A) were excised and analyzed by LC-MS/MS[17]. For in vivo Ser-171 phosphorylation, OST1/SnRK2.6-GFP was transiently expressed in *Arabidopsis* mesophyll cell protoplasts. The protoplasts were incubated with or without 20 μM ABA for 15 min, and OST1/SnRK2.6 proteins were purified by immunoprecipitation using anti-GFP antibodies. After SDS-PAGE and CBB staining, OST1/SnRK2.6-GFP bands were excised and analyzed by LC-MS/MS[17].

**Analysis of stomatal ABA response.** Infrared-based gas exchange analyzer systems were used including an integrated Multiphase Flash Fluorometer (Li-6800-01A or Li-6400; Li-Cor Inc.) for gas exchange analyses. Plants were grown on soil in Percival growth cabinets at a 12/12 h, 21 °C/21 °C day/night cycle, a photosynthetic photon flux density of ~90 mmol m$^{-2}$ s$^{-1}$, and 70–80% relative humidity for 6–7 weeks. Mature rosette leaves were detached at the basal part of petiole by a razor blade, and re-cut twice under distilled and deionized water. The petioles of the leaves were then immersed in ddH$_2$O for gas exchange analysis. The detached leaves were clamped and the environment of the leaf chamber was controlled at 400 ppm ambient CO$_2$, 23–24 °C, ~65% relative air humidity, 150 μmol m$^{-2}$ s$^{-1}$ photon flux density, and 500 μmol s$^{-1}$ flow rate until stomatal conductance stabilized. One or 2 μM ± -ABA was applied to the petiole for kinetic stomatal conductance response analyses as described[59].

**Patch-clamp analyses.** Guard cell protoplasts from 4 to 6-week-old Arabidopsis plants were prepared[24,33]. ABA-activated S-type anion channel current recordings were carried out by using an Axon 200 A amplifier (Axon instruments) and a Digidata 1440 A low-noise data acquisition system. Epidermal tissues were isolated from one or two rosette leaves and collected using a nylon mesh (100-μm pore size). Subsequently the epidermal tissues were incubated in 10-ml protoplast isolation solution containing 500 mM D-mannitol, 1% cellulase R-10 (Yakult Pharmaceutical Industry), 0.5% macerozyme R-10 (Yakult Pharmaceutical Industry), 0.5% bovine serum albumin, 0.1% kanamycin sulfate, 0.1 mM CaCl$_2$, 0.1 mM KCl, and 10 mM ascorbic acid for 16 h at 25 °C on a circular shaker at 50 rpm. Guard cell protoplasts were collected through a nylon mesh (10-μm pore size) and then washed two times with protoplast suspension solution containing 500 mM D-sorbitol, 0.1 mM CaCl$_2$, and 0.1 mM KCl (pH 5.6 with KOH) by centrifugation (200 × g for 5 min at room temperature). Isolated guard cell protoplasts were stored on ice before use.

S-type anion currents in guard cell protoplasts were recorded using the whole-cell patch-clamp technique[24,33]. The pipette solution was composed of 150 mM CsCl, 2 mM MgCl$_2$, 5.86 mM CaCl$_2$, 6.7 mM EGTA, and 10 mM Hepes-Tris (pH 7.1). 5 mM Mg-ATP was added to the pipette solution freshly before use. The bath solution was composed of 30 mM CsCl, 2 mM MgCl$_2$, 1 mM CaCl$_2$, and 10 mM MES-Tris (pH 5.6). Osmolalities of the pipette solution and the bath solution were adjusted to 500 mosmol kg$^{-1}$ and 485 mosmol kg$^{-1}$ using D-sorbitol, respectively. In Fig. 3, guard cell protoplasts were pre-incubated for 20 min in the bath solution containing 50 μM ABA prior to recordings, and ABA was added to the pipette solution. In Supplementary Fig. 10, guard cell protoplasts were pre-incubated for 30 min in the bath solution containing 10 μM ABA prior to recordings.

**Two-electrode voltage clamp recordings.** The PCR amplified cDNA fragments of OST1, SLAC1, PYL9/RCAR1, ABI1, M3Kδ1, M3Kδ6, and M3Kδ7 were cloned into the oocyte expression vector pNB1 by using an advanced uracil-excision based cloning strategy as previously described[60]. The mutant isoforms OST1-S171A, M3Kδ6-K775W, and M3Kδ7-K740W were generated using the Quikchange Site-Directed Mutagenesis kit (Agilent Technologies). Linearized plasmids were used to generate cRNAs via the mMESSAGE mMACHINE® T7 kit (Thermo Fisher Scientific, Catalog number: AM1344). Surgically extracted ovaries of *Xenopus laevis* were ordered from Nasco (Fort Atkinson, Wisconsin, product number: LM00935) and Ecocyte Bio Science US (Austin, Texas) and oocytes were isolated as previously described[61]. Five nanogram of cRNA of each construct *OST1, OST1-S171A, SLAC1, PYL9/RCAR1, ABI1* and 0.5 ng cRNAs of each construct *M3Kδ1, M3Kδ6, M3Kδ7, M3Kδ6-K775W, M3Kδ7-K740W* were co-injected into isolated oocytes in the indicated combinations. Oocytes were incubated at 16 °C for 2 days in ND96 buffer (1 mM CaCl$_2$, 1 mM MgCl$_2$, 96 mM NaCl, 10 mM MES/Tris, pH = 7.5). Osmolarity was adjusted to 220 mosmol kg$^{-1}$ by D-sorbitol. Using a Cornerstone (Dagan) TEV-200 amplifier and a Digidata 1440 A low-noise data acquisition system with pClamp software (Molecular Devices), two-electrode voltage clamp recordings were performed in a bath solution containing 1 mM CaCl$_2$, 2 mM KCl, 24 mM NaCl, 70 mM Na-gluconate, 10 mM MES/Tris, pH 7.4, Osmolarity was

adjusted to 220 mosmol kg$^{-1}$ by D-sorbitol. ABA was injected into oocytes to achieve a final concentration of 50 μM for analyses of ABA activation of SLAC1 currents. Steady-state currents were recorded with 3 s voltage pulses ranging from +40 mV to – 120 mV in −20 mV decrements, followed by a "tail" voltage of −120 mV and the holding potential was kept at 0 mV.

SLAC1-mediated currents in oocytes vary showing either time-dependent relaxation or more instantaneous currents when using a chloride bath solution[61,62]. Furthermore, ion channel activities display different magnitudes from one oocyte batch to another due to protein expression level variation among batches of oocytes. To avoid time-of-measurement and inter-batch dependence in the data, H$_2$O-injected control and other indicated controls were included in each batch of oocytes and control experiments were recorded intermittently with the investigated conditions. Data from one representative oocyte batch are shown from the same batch in each figure panel and at least three independent batches of oocytes were investigated and showed consistent findings.

**Mesophyll cell protoplast assays**. Mesophyll cell protoplasts were isolated as described previously[63] from 3-4-week-old *Arabidopsis* leaves. 10–20 μg of pUC18 plasmids carrying *35 S:OST1/SnRK2.6-GFP:nosT* or *35 S:M3Kδ6-FLAG:nosT* and 30 μg protoplasts were used for 20% PEG-mediated transient expression. After overnight incubation in incubation buffer (10 mM MES-KOH pH 6.0, 0.4 M mannitol, 20 mM KCl, 1 mM CaCl$_2$), protoplasts were incubated in 10 μM ABA or 0.8 M mannitol or in control buffer for 15 min and harvested by centrifugation at 13,000 × *g* for 1 min. After the supernatants were removed, 20 μL SDS-PAGE loading buffer was added and incubated at 95ºC for 3 min.

**Measurements of leaf temperatures by thermal imaging**. Plants grown 4–5 weeks on soil were sprayed with 20 μM ABA dissolved in water. After 3 h under white light in the growth room, images were captured using an infrared thermal imaging camera (T650sc; FLIR, Wilsonville, Oregon). Leaf temperatures were determined as average temperatures of each whole leaf area by using Fiji software (ImageJ version: 2.0.0-rc-59/1.51n).

**Creating CRISPR/Cas9-based knockout *Arabidopsis***. The *m3kδ1* and *m3kδ7* CRISPR/Cas9 deletion knockout mutants were generated using CRISPR/Cas9 gene editing technology[64–66] in the *m3kδ6-2* mutant background. We used two guide RNAs to generate a large deletion in each target gene. The target sequences in *M3Kδ1* were TACGGAAGCTCCACATCGGCGG and GATGCAAGTCGTTGG AGCTGTGG (PAM sites are underlined). Targets for *M3Kδ7* were GACGGAG TTCCAGATCTCCGGG and CCAGAGAGCAGCAGTTCCCAGT.

The designed *m3kδ1crispr* mutants were genotyped with the primer pair Delta1-GT1 and Delta1-GT2, which would generate a fragment of about 750 bp when the designed deletion took place. The primer pair could not amplify WT genomic DNA due to the large size of the fragment. To determine zygosity of *m3kδ1crispr* mutants, we used the primer set Delta1-GT1 + Delta1-GT3, which amplifies a 777 bp fragment from WT DNA, but could not amplify a band in a homozygous mutant.

For *m3kδ7crispr* mutants, we used Delta7-GT1 and Delta7-GT4, which would generate a fragment of about 1390 bp if mutant DNA is used as PCR template. The primer pair could not amplify WT DNA because of the large fragment size. The Delta7-GT1 and Delta7-GT3 primer pair was able to generate a fragment of 1125 bp when WT DNA was used as PCR template. The Delta7-GT1/GT3 was used to differentiate homozygous *m3kδ7crispr* mutants from heterozygous *m3kδ7crispr* mutants. After isolating homozygous *m3kδ1crispr m3kδ6-2* and *m3kδ7crispr m3kδ6-2* mutants, these lines were crossed and homozygous triple mutants were recovered in the T2 generation. Primers for genotyping: Delta1-GT1: 5′-TTGTTGGTTCCACGAACGGA-3′, Delta1-GT2: 5′-GATGGCCGTAAAT GCGGTTC-3′, Delta1-GT3: 5′-CGGATCAGGATCAGAGACGC-3′, Delta7-GT1: 5′-TGCATAAGGTGGTGAGCGAA-3′, Delta7-GT3: 5′-CCAAACCCTGCA TCCCAGAT-3′, Delta7-GT4: 5′-GTCAAGGAAGAAGCGACCCA-3′

**Creating amiRNA knock-downs targeting *M3Kδ1, δ6* and *δ7***. The amiRNA sequence was designed using the WMD3 (http://wmd3.weigelworld.org/cgi-bin/webapp.cgi) and PHANTOM database (http://phantomdb.ucsd.edu). The amiRNA containing the target sequence (5′-TACGACTTGCATCGGGTTCAA-3′) for *M3Kδ1, M3Kδ6*, and *M3Kδ7* was amplified by PCR using primers (I: 5′-gaTACG ACTTGCATCGGGTTCAAtctctctttttgtattcc-3′, II: 5′-gaTTGAACCCGATGCAAG TCGTAtcaaagagaatcaatga-3′,

III: 5′-gaTTAAACCCGATGCTAGTCGTTtcacaggtcgtgatatg-3′,

IV: 5′-gaAACGACTAGCATCGGGTTTAAtctacatatatattcct-3′), and inserted into the vector pFH0032[21]. *Arabidopsis* (Col-0) plants were used for floral-dip transformation. Three independent homozygous T3 seeds were used in the seed germination assays.

**BiFC analyses**. Constructs for BiFC analyses were generated by ligation of coding sequences of ABI1, RopGEF1, OST1/SnRK2.6, SnRK2.2, SnRK2.4, SnRK2.10, M3Kδ6, and M3Kδ7 into pSPYCE(M) or pSPYNE173 using the USER Cloning technology (see Supplementary Table 1 for primer sequences). Plasmids were transformed into *Agrobacterium tumefasciens* (GV3101) and co-infiltrated with a plasmid expressing the silencing suppressor p19 in leaves of 6-week-old *Nicotiana benthamiana* plants. Subcellular localization analyses were performed using a Nikon Eclipse TE2000-U confocal microscope. Images were acquired using a Plan Apo VC 60XA/1.20 WI objective using identical settings (exposure time and gain). Three independent experiments were conducted where three leaves were analyzed for each combination. 5 z-stacks were acquired for each leaf. Maximum projections of z-stacks for each BiFC combination were quantified using Fiji and normalized over an infiltration control expressing p19 only.

**Statistics**. Cotyledon greening assays were analyzed by two-way ANOVA followed by Tukey's tests. Leaf temperatures were analyzed by one-way ANOVA followed by Tukey's tests.

**Reporting summary**. Further information on research design is available in the Nature Research Reporting Summary linked to this article.

## Data availability
*Arabidopsis* mutants and transgenic lines used in this study are available upon request from the corresponding author. The source data for Figs. 1, 3–6, Supplementary Figs. 7–11, 13, 15 and 16 are provided as a Source Data file.

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

## Acknowledgements

This research was funded by National Institutes of Health (GM060396-ES010337) and National Science Foundation (MCB-1900567) grants to J.I.S., a Postdoctoral Fellowship for Research Abroad from the Japan Society for the promotion of Science to Y.T., a Ciencias sem Fronteiras/CNPq fellowship (203406/2014-1) to P.H.O.C. and a scholarship from the China Scholarship Council to L.Z., an EMBO long-term post-doctoral fellowship (ALTF334-2018) to G.D., and in part NIH grant GM114660 to Y.Z. We thank Dr. Majid Ghassemian (University of California, San Diego) for advice at analyzing LC-MS/MS data (National Institutes of Health grant number: S10OD021724). We thank Mr.

Brian Chang and Ms. Katie H. Lee (University of California, San Diego) for help with amiRNA lines. We thank Ms. Krystal Bosmans and Mr. Thien Trac (University of California, San Diego) for help with producing recombinant proteins.

## Author contributions

Y.T. and J.I.S. conceived of the project; Y.T., J.Z, P.H., F.H., and J.I.S. designed research; Y.T., J.Z., P.H., P.H.O.C., L.Z., G.D., S.M., C.G., and F.H. performed experiments; C.G. and Y.Z. generated CRISPR/Cas9 plants; Y.T., J.Z., P.H., P.H.O.C., L.Z., G.D., S.M., C.G., Y.Z., F.H., and J.I.S. analyzed data; and Y.T., J.Z., P.H., F.H., and J.I.S. wrote the manuscript.

## Competing interests

The Authors declare the following competing interests: The University of California San Diego has submitted a patent (US patent 2/816,492, pending) on behalf of Y.T., J.I.S., and F.H. on M3K-dependent modulation of abscisic acid signaling and osmotic stress-linked responses. The other authors declare no competing interests.
