## [Peer Review File · Nature Communications]

Reviewers' comments:

Reviewer #1 (Remarks to the Author):

Takahashi et al. report the identification of 3 subgroup B3 M3Ks that are important for ABA responses and for fast osmotic stress responses. They provide strong evidence that the requirement for ABA responses is due to phosphorylation of S171 in SnRK2.6/Ost1 and likely the corresponding residues in SnRK2.2 and SnRK2.3. This is a conceptual breakthrough and should be published with priority under the condition that the authors can satisfactorily address the following points:

1. In Fig 1d, the mobility of Ost1 is ~45 kDa, in Fig. 1g the mobility His6-Ost1 is above 50 kDa. Where does this discrepancy come from?
2. SnRK2.2, 2.3, and 2.6 have partially redundant functions in ABA responses. It would further strengthen the conclusions of the paper if the authors could generate SnRK2.6 S171A-corresponding non-phosphorylatable mutations in SnRK2.2 and 2.3 and test whether in the triple non-phosphorylatable mutant ABA responses are reduced to a similar level as in a SnRK2.2/2.3/2.6 deletion mutation, as predicted by the presented model.
3. The triple d1/d6/d7 mutant only mildly affects ABA induction and SnRK2 in-gel kinase activity, suggesting that additional M3Ks can phosphorylate the 3 ABA responsive SnRK2s. The same triple mutant strongly reduced rapid ABA-independent osmotic stress responses and at least M3K d1 can induce weak in-gel kinase activity in one of the fast osmotic stress-mediating SnRK2s. The simplest model would therefore be that i) M3Ks induce osmotic stress responses by direct phosphorylation of SnRK2s, and ii) that d1, d6, and d7 are the major M3Ks to do so, at least to a larger degree than phosphorylating the 3 ABA-responsive SnRK2s. Could you test the ability of d6 and d7 to phosphorylate SnRk2.4 and possibly other fast osmotic stress-mediating SnRK2s?
4. Fig. 1e: Please provide full gel images in supplemental data rather than very narrow gel slices.
5. In vitro reconstitution: please indicate molar amounts instead of μg amounts. Reconstitution as well as the mechanisms of SnRK2 inhibition by HAB1 depend on stoichiometries, which are immediately clear when molarities are used.

Reviewer #2 (Remarks to the Author):

This work by Takahashi et al. proposed that B group Raf-like M3K regulates activation of SnRK2 kinases in response to ABA or osmotic stress. The authors identified M3Ks by amiRNA screening, and demonstrated that M3Ks can phosphorylate SnRK2 S171, which is a critical site for SnRK2 activity. M3Ks are also involved in the regulation of a slow anion channel SLAC1, suggesting their functional roles in guard cells. Furthermore, T-DNA knockout or CRISPR-based M3K disruptants showed ABA-insensitive phenotype in hypocotyl greening assay. Based on those results, authors concluded that M3Ks are a missing component of the early ABA signaling in plants. Overall, experiments were well designed, and data quality is excellent. Their findings could bring us new insights to ABA or osmotic stress signaling in higher plants. However, there are significant problems as follows.

[Major points]

1. In this manuscript, authors described about 1) identification of M3Ks as upstream regulators of SnRK2, 2) M3Ks directly phosphorylates SnRK2, 3) phenotyping of M3K disruptants, 4) involvement of M3Ks in osmotic stress signaling. However, those points are overlapped with

Saruhashi et al. PNAS (2015), in which ARK, a B group Raf-like M3K, was identified as an upstream regulator of SnRK2 in *Physcomitrella patens*. Therefore, the authors should reform their manuscript to focus on the differences between Arabidopsis M3Ks and *Physcomitrella* ARK.

2. In Fig.1d, SnRK2 activity is not completely dismissed in *ami_m3k*. The same with Fig. 6a-d. What does this mean? Some other M3Ks or kinases could be involved in the regulation of SnRK2?

3. In Fig. 2 and 3, the authors used a mutant form of SnRK2 (S171A) for kinase assay or complementation of *ost1*. However, the presented data is not enough to confirm the biological significance of S171 phosphorylation in ABA signaling. It is highly possible that SnRK2 S171A lost its enzymatic activity, because S171 is located on the kinase activation loop. If so, it is not surprising that SnRK2 S171A did not complement with *ost1* phenotype. Authors should check that some other mutations in kinase activation loop can complement *ost1*.

4. The authors concluded that M3K phosphorylates Ser171 on SnRK2 in response to ABA. However, there is no evidence of *in vivo* S171 phosphorylation by M3K. It is necessary to check the phosphorylation status of Ser171 in the *m3k* disruptants, for example, MS analysis of immunoprecipitated SnRK2 from WT and *m3k* disruptants, or western blot analysis using an anti-pS171 antibody.

5. There is no evidence to show *in vivo* interaction between SnRK2 and M3K.

6. Phenotyping of *m3k* disruptants in Fig. 5. Why don't you check some other ABA responses in the mutant? For example, stomatal conductance, root elongation, or gene expression. Especially, authors demonstrated that M3Ks are involved in the regulation of SLAC1 activity in Fig. 4, suggesting that M3Ks could have some roles in guard cells. This point needs to be confirmed in *planta* using M3K disruptants. In summary, it is still unclear whether M3Ks are global regulators of ABA signaling in higher plants or not.

Reviewer #3 (Remarks to the Author):

This submission reports identifying the upstream re-activators of the core SnRK2 kinases in ABA signaling. These activators were revealed as mutations/mutants in the M3K δ s (MAPKK kinases) family generated by *ami*-RNA that showed reduced ABA sensitivity on seed germination. Allelic mutations in some of these M3K δ 6/7 also showed little or no detectable "SnRK2" activity upon ABA or osmotic treatments.

The authors have provided several lines of evidence to support their hypothesis, using OST1/SnRK2.6 as the representative SnRK2. The SnRK2 kinase OST1/SnRK2.6 expressed in *E. coli* is active (unlike other members), presumably by "autophosphorylation". But once OST1/SnRK2.6 is inactivated by protein phosphatases (e.g. by 'non-specific' CIP or specific HAB), the SnRK2 cannot resurrect itself by auto-phosphorylation. Then, how does OST1/SnRK2.6 gets re-activated? The answer is that this needs upstream kinases, namely the M3Ks δ . The supporting lines of evidence are the following:

(A) In-gel kinase assays, the protein extracts from the M3K δ mutants (extracts pooled from mutants? I suppose) have reduced activities for at least three kinases, which the authors attributed to SnRK2s.

(B) OST1/SnRK2.6 treated by Calf Intestinal Phosphatase was unable to autoactivate a second time, but needs trans-phosphorylation by M3Ks δ 1/ δ 6/ δ 7 on S171, S175 and Thr176 (Fig. 2C). The analyses then concentrated on the S171 site (based on the authors' evidence that it is not autophosphorylated, Fig. 2d).

(C) Functional complementation of the *ost3-1* mutant and ABA-activation of S-type anion channel currents by patch-clamp using guard cell protoplasts from these stably transgenic lines support the importance of S171 for OST1/SnRK2.6 kinase activity (Fig. 2e; Fig. 3).

(D) In vitro reconstitution of the ABA pathway showing that the addition of M3K δ 6 restored the OST1/SnRK2.6 kinase activity (using phosphorylation of a transcription factor as the readout), previously deactivated by the HAB1 protein phosphatase 2C. This is supplemented by reconstitution of the pathway in *Xenopus* oocytes, with the activation of anion channel SLAC1, as the readout.

(E) Disruptions in double or triple mutants of M3Ks δ 1/ δ 6/ δ 7 lead to ABA-insensitive seed germination, as compared to the wild type, based on cotyledon greening.

The results are very enlightening and add to previous pharmacological studies that this family of SnRK2s are activated by different upstream mechanisms. One of these is thus via the mitogen-activated kinase pathways that are activated by ABA-dependent and -independent signals.

May I make a couple of remarks that might improve the weight of the evidence and the clarity of the paper?

Major point:

1. Figure 3 showed the functional importance of S171 in the OST1 protein by complementation of the *Arabidopsis* *ost1-3* mutant phenotype (transpiration, S-type anion currents) using either the WT transgene or the counterpart with the S171A mutation. This is fine as a prove of principle. But to connect the functional significance of S171 better with the upstream M3K δ s, and in planta, have the authors attempted to by-pass the M3K δ mutations with the transgene OST1/SnRK2.6 bearing a phospho-mimic Ser171?

2. The phenotypic characterization for the M3K δ mutants could be more complete. Because the *ost1-3* mutant showed an excessive transpiration phenotype, one would expect that the upstream M3K δ to have comparable stomatal dysfunction, assuming that OST1/SnRK2.6 and M3K δ function in the same pathway in planta. If these M3K δ have several SnRK2 targets, have the authors actually observed that the stomatal phenotype in M3K δ triple mutant being even more severe than that in *ost1-3*. (Fujii et al., 2011). More data on stomatal phenotypes and patch-clamp analysis on the M3K δ mutant guard cells should add more solidity to the in planta results.

Minor queries:

3. I find that the description on mutant isolation not easy to follow. It is not intuitive to relate the 14 mutants back to the three lines that were included in the prior collection of 25 mutants. Are they independent or some/all are from the same parental lines etc? For this reviewer, because a mutational event in a specific locus is mathematically rare in an unbiased screen, many progeny from the same parental lines could like be from the same mutant (for a given phenotype), unless proven otherwise.

4. Fig. 1d, *ami_m3K*, for the in-gel kinase assays. Are the phosphorylation activities from pooled mutant seedlings of the genotypes M3K δ 5, M3K δ 7, M3K δ 1, M3K δ 6, M3K δ -CTR1? Or from a multiple mutant?

5. From the above in-gel kinase assays alone, besides the mol. weights of the kinase activities as clues, a non-expert would unlikely jump to the conclusion that the reduced histone

phosphorylation signals must correspond to the SnRK2.2, 2.3 and 2.6. Can the authors explain this better? I think histones are also used as a surrogate substrate for many kinases.

6. Fig. 6 comes back to the mutant M3K δ characterization by osmotic stress and ABA. These results maybe better if presented earlier along with Fig. 1d. If the figure is too dense as a consequence, some of the results could be transferred to the supplementary data section.

7. Besides the one (and older) reference 19 cited in this manuscript, there are two other recent references on SnRK2.4, being targets of the ABA-related PP2Cs (Krzywinska et al., 2016b; Krzywinska et al., 2016a). There is also a paper by Danquah et al. (2015) that reported an ABA-activated MAP kinase cascade. The authors may want to check them out to see whether they may be worth citing in the same vein of ABA-dependent and ABA-independent signal cross-talk.

8. In Fig. 5d, Fig. 6a, b. Some mutants are designated only by crispr; is this the most severe multiple mutant or a pool of mutants created by crispr? Please clarify how cotyledon "greening" assays shown in the panels (e) (h) and (j) were quantified. I suppose that this is purely visual, to give a rough indications of the mutant phenotypes on germination. Do the authors think that it might be useful to have quantification of chlorophyll for a given number of germinated plants etc to give some precisions? It appears that some mutant lines are greener than others, and the sizes of the different mutants are different. But this kind of cross comparisons are difficult because these lines were germinated on different concentrations of exogenous ABA, for different lengths of time. Do these M3K mutants display altered seed dormancy (reduced?) as a quantitative indicator of sensitivity to endogenous ABA?

References:

Danquah, A., de Zelicourt, A., Boudsocq, M., Neubauer, J., Frei Dit Frey, N., Leonhardt, N., Pateyron, S., Gwinner, F., Tamby, J.P., Ortiz-Masia, D., Marcote, M.J., Hirt, H., and Colcombet, J. (2015). Identification and characterization of an ABA-activated MAP kinase cascade in *Arabidopsis thaliana*. *Plant J.* 82, 232-244.

Fujii, H., Verslues, P.E., and Zhu, J.-K. (2011). *Arabidopsis* decuple mutant reveals the importance of SnRK2 kinases in osmotic stress responses *in vivo*. *Proc Natl Acad Sci U S A* 108, 1717-1722.

Krzywinska, E., Kulik, A., Bucholc, M., Fernandez, M.A., Rodriguez, P.L., and Dobrowolska, G. (2016a). Protein phosphatase type 2C PP2CA together with ABI1 inhibits SnRK2.4 activity and regulates plant responses to salinity. *Plant Signaling & Behavior* 11, e1253647.

Krzywinska, E., Bucholc, M., Kulik, A., Ciesielski, A., Lichocka, M., Debski, J., Ludwikow, A., Dadlez, M., Rodriguez, P.L., and Dobrowolska, G. (2016b). Phosphatase ABI1 and okadaic acid-sensitive phosphoprotein phosphatases inhibit salt stress-activated SnRK2.4 kinase. *BMC Plant Biol.* 16, 136.

Reviewer #4 (Remarks to the Author):

Takahashi et al's study firmly places MAP3Ks as a core ABA/osmotic signalling component in *Arabidopsis thaliana*. ABA signalling has been shown by various groups to involve ABA binding to the PYR/PYL/RCAR which inhibits a PP2C phosphatase releasing a SnRK2 kinase to interact/phosphorylate downstream targets. Here the authors find a new component in this pathway. The authors conclude that 3 MAP3K δ 1/6/7 are required to fully activate SnRK2 kinases for complete ABA signalling to occur and that these kinases are likely to represent a convergence point between (ABA independent) osmotic responses and ABA signalling. This

component has been previously overlooked due to the nature of in vitro studies which spontaneously phosphorylate SnRK2s. To resolve the effects authors had to dephosphorylate the SnRK2 to show that 1) they are not autophosphorylated and 2) that specific MAP3K can phosphorylate SnRK2s.

This is a follow up to the Schroeder lab's pioneering 2013 Plant Cell paper where they introduce a screen that identifies plants containing amiRNA can be used to overcome 'redundancy' and identify molecular components involved in plant function. In that work they find plants with amiRNA targeting several MAP3Ks display ABA-insensitive germination compared to wildtype. Here, they fully validate that preliminary observation and go on to show that some of these MAP3K interact with OST1 (SnRK2.6) and phosphorylate S171 (not previously known for OST1 activation) and this is required for ABA induced stomatal closure via activation of SLAC1. Furthermore, they show evidence that MAP3K are also involved in osmotic induced phosphorylation of SnRK2s.

The study is very high quality and I find little fault with the experimental approaches and results as presented. The finding that MAP3Ks likely act as a convergence point for ABA and osmotic signalling is significant.

Major points to address

1) Shitamichi et al 2013 Plant Biotechnology showed that MAP3Kdelta4 overexpression had the same relief of ABA inhibition of germination as the MAP3Kdelta triple knockouts here, it would be appropriate to cite and discuss. Furthermore, Li et al 2015 PCP also showed that germination of Raf10 and 11 (B2 MAP3Ks incl. MAP3Kdelta4) are insensitive to ABA and overexpression plants are more sensitive to ABA. Again this paper is not cited. There are also several more MAP3K papers published from Japan that show involvement of MAP3Ks in ABA signalling albeit not showing the likely interaction with SnRK2s.

2) This manuscript contains three distinct but related elements to fully demonstrate the involvement of MAP3K in ABA/osmotic signalling. Firstly, the examination of OST1 regulation of SLAC1-facilitated stomatal closure, then the ABA inhibition of germination, and finally osmotic activation of SnRK2s. All point to the involvement of MAP3K. These findings highlight the importance of MAP3K but none of these aspects of the study are fully mature. For instance:

A) what is MAP3K interacting with in roots during the germination responses to ABA – do they involve SnRK2s?

B) What are the ABA-related stomatal phenotypes of the MAP3K mutants? i.e are the MAP3K essential to ABA-activation of SLAC1 (as carried out in vitro).

C) What are the plant osmotic related phenotypes in the map3kdelta triple knockouts?

Whilst it cannot be debated that MAP3K are clearly identified as a component in ABA signalling, the firmness of conclusions regarding the exact mechanisms of how they are involved needs several dotted lines of evidence joining. This is especially true as this study convincingly shows that in vitro oocyte studies do not to fully represent what occurs in planta. Therefore, can the authors fully recapitulate the findings of the in vitro experiments carried out here in planta? I would have thought this to be very important to support the conclusions? Alternatives (not required – but suggestions) could also be is ost1 redundant in map3kdelta triple knockout lines? or can the map3kdelta triple knockout phenotypes be rescued by an activated OST1? Or potentially are there confirmatory phenotypes of overexpression plants?

3) On that note. In vitro phosphorylation studies may force phosphorylation to occur when it doesn't in the cell. Can the authors show co-expression data/in vivo interaction data as further evidence of the interaction, or do they think the genetic evidence is currently sufficient?

4) What are the phenotypes of the single and double map3kdelta T-DNA knockouts?

Other points for consideration

In Fig 1 g, "M3Kdelta1" was labelled twice in the upper panel; Statistical analysis is missing in Fig 1e, 4f, 6b and d.

Only M3Kdelta6's data is often been presented not M3Kdelta1 and M3Kdelta7 in several biochemical analysis (such as Fig2 a-b, Fig4 a-c). It's fascinating to see M3Kdelta6 can directly phosphorylate inactive OST1 (D140A), does this also apply to the other two M3Ks? It would be ideal to specify whether M3Kdelta1, 6 and 7 share the same regulation mechanisms in phosphorylating SnRK2s. In Fig 6e, M3Kdelta1 activates SnRK2.2 and 2.3, does this suggest M3Ks have distinctive SnRK2 substrate preference, does this have implications for downstream targets and distinctive roles?

In Fig 4d, upon co-injection with PYL9+ABI1+OST1+SLAC1, M3Kdelta6 was able to slightly activate SLAC1 anion activity without ABA by eliciting greater inward currents compared to PYL9+ABI1+OST1+SLAC1+M3Kdelta1. In contrast, in Supp Fig 5, all three M3Kdelta1, 6 and 7 were able to activate SLAC1 in a similar manner by interacting with OST1. Can the authors explain this data? Do they think a set of negative control, such as co-expression dephosphorylated OST1 with M3Ks +SLAC1 (OST1S175A+ M3Kdelta6+ SLAC1) would further validate that M3Ks can robustly re-activate dephosphorylated OST1? This could clarify which dephosphorylated residue in OST1

can be fully activated by M3Ks, as authors have mentioned that multiple putative phosphorylation sites have been identified in OST1?

Does the word core need to be in the title? – it doesn't quite scan easily at present

Reviewer #1 (Remarks to the Author):

Takahashi et al. report the identification of 3 subgroup B3 M3Ks that are important for ABA responses and for fast osmotic stress responses. They provide strong evidence that the requirement for ABA responses is due to phosphorylation of S171 in SnRK2.6/Ost1 and likely the corresponding residues in SnRK2.2 and SnRK2.3. This is a conceptual breakthrough and should be published with priority under the condition that the authors can satisfactorily address the following points:

1. In Fig 1d, the mobility of Ost1 is ~45 kDa, in Fig. 1g the mobility His6-Ost1 is above 50 kDa. Where does this discrepancy come from?

Response:

This difference in mobility results from the 6xHis-tag (e.g. Figure 1g) and other additional sequences including thrombin and enterokinase cleavage sites and restriction enzyme sites in the pET-30a(+) vector that we used for *E.coli* expression of OST1/SnRK2.6 in Figure 1g. We have now added information to the Methods section clarifying this point:

“Note that the mobilities of recombinant and transgenic proteins in the present study depended on tags. For example, the pET-30a(+) vector used for *E.coli* expression of OST1/SnRK2.6 in figures 1g, 4a and b includes a 6xHis-tag and other sequences including thrombin and enterokinase cleavage sites and restriction enzyme sites”

2. SnRK2.2, 2.3, and 2.6 have partially redundant functions in ABA responses. It would further strengthen the conclusions of the paper if the authors could generate SnRK2.6 S171A-corresponding non-phosphorylatable mutations in SnRK2.2 and 2.3 and test whether in the triple non-phosphorylatable mutant ABA responses are reduced to a similar level as in a SnRK2.2/2.3/2.6 deletion mutation, as predicted by the presented model.

Response:

Thank you for this suggestion to test whether SnRK2.2/2.3 carrying a non-phosphorylatable mutation corresponding to the OST1 S171A affects kinase activation. Indeed, we found conserved serine residues in SnRK2.2 (Ser-180) and SnRK2.3 (Ser-172). However, to create stable homozygous transgenic plants which have these point-mutations in each SnRK2 kinase gene may take more than one year. Among other reasons, transformation of the *snrk2.2/2.3/2.6* triple mutant is not very easy, due to the fragile nature of these plants. Transforming and crossing would take a substantial time. To address this comment, we pursued mesophyll cell protoplast transient gene expression. We expressed SnRK2.2(S180A) and SnRK2.3(S172A), and found that these kinases are not activated by ABA using *in vitro* phosphorylation assays in contrast to their respective wild-type isoforms. We included this result in Supplementary Figure 14b in the revised manuscript.

3. The triple d1/d6/d7 mutant only mildly affects ABA induction and SnRK2 in-gel kinase activity, suggesting that additional M3Ks can phosphorylate the 3 ABA responsive SnRK2s. The same triple mutant strongly reduced rapid ABA-independent osmotic stress responses and at least M3K d1 can induce weak in-gel kinase activity in one of the fast osmotic stress-mediating SnRK2s. The simplest model would therefore be that i) M3Ks induce osmotic stress responses by direct phosphorylation of SnRK2s, and ii) that d1, d6, and d7 are the major M3Ks to do so, at least to a larger degree than phosphorylating the 3 ABA-responsive SnRK2s. Could you test the ability of d6 and d7 to phosphorylate SnRk2.4 and possibly other fast osmotic stress-mediating SnRK2s?

Response:

Thank you for your suggestion. To test the ability of M3K δ 6 and δ 7 to phosphorylate SnRK2.4, we have been trying to produce kinase inactive GST-SnRK2.4 and GST-SnRK2.10 proteins in *E.coli*. However, several attempts have failed to produce the mutant proteins possibly due to protein instability, in contrast to the OST1/SnRK2.6 mutant isoform (Fig. 2b). For this reason, it is difficult to examine whether M3K δ 6 and δ 7 have an ability to phosphorylate SnRK2.4/2.10. However, we have shown in *in-gel* kinase assays that the wild type isoform of SnRK2.4 is activated in the presence of M3K δ 1 (Fig. 6e).

4. Fig. 1e: Please provide full gel images in supplemental data rather than very narrow gel slices.

Response:

We prepared Supplementary Figure 17 showing full gel images.

5. In vitro reconstitution: please indicate molar amounts instead of μ g amounts. Reconstitution as well as the mechanisms of SnRK2 inhibition by HAB1 depend on stoichiometries, which are immediately clear when molarities are used.

Response:

Thank you for this helpful suggestion. We have revised the Methods section and provide molecular amounts instead of μ g amounts (Page 9, line 359).

Reviewer #2 (Remarks to the Author):

This work by Takahashi et al. proposed that B group Raf-like M3K regulates activation of SnRK2 kinases in response to ABA or osmotic stress. The authors identified M3Ks by amiRNA screening, and demonstrated that M3Ks can phosphorylate SnRK2 S171, which is a critical site for SnRK2 activity. M3Ks are also involved in the regulation of a slow anion channel SLAC1, suggesting their functional roles in guard cells. Furthermore, T-DNA knockout or CRISPR-based M3K disruptants showed ABA-insensitive phenotype in hypocotyl greening assay. Based on those

results, authors concluded that M3Ks are a missing component of the early ABA signaling in plants.

Overall, experiments were well designed, and data quality is excellent. Their findings could bring us new insights to ABA or osmotic stress signaling in higher plants. However, there are significant problems as follows.

[Major points]

1. In this manuscript, authors described about 1) identification of M3Ks as upstream regulators of SnRK2, 2) M3Ks directly phosphorylates SnRK2, 3) phenotyping of M3K disruptants, 4) involvement of M3Ks in osmotic stress signaling. However, those points are overlapped with Saruhashi et al. PNAS (2015), in which ARK, a B group Raf-like M3K, was identified as an upstream regulator of SnRK2 in *Physcomitrella patens*. Therefore, the authors should reform their manuscript to focus on the differences between *Arabidopsis* M3Ks and *Physcomitrella* ARK.

Response:

Thank you for this suggestion. We had cited in the original manuscript, MAP3K δ 1, δ 6 and δ 7 have homologies to *Physcomitrella* ARK kinase which was reported to phosphorylate *Physcomitrella* SnRK2. To address this comment, we have now further summarized the ARK findings in the Results section ("Note that a *Physcomitrella patens* protein kinase ARK showing similarity to these M3Ks was recently reported to phosphorylate a *Physycomitrella* SnRK2 kinase²⁸", Page 3, line 104) and added further discussion on the ARK kinase ("Recent studies show that ARK kinase is required for *Physcomitrella* ABA and drought stress responses including phosphorylation of transcription factors through SnRK2 kinases^{38,39}.", Page 7, line 268; "The M3K ARK is required for osmotic stress tolerance in *Physcomitrella*^{24,39}.", Page 7, line 281). In addition to the identification of the MAP3Ks, our manuscript provides new findings as follows.

1) We identify and characterize a new *Arabidopsis* MAPKK kinase family clade through a combination of amiRNA-based redundancy-circumventing forward genetics and several functional analyses. Our research demonstrates that these MAPKK kinases are a new regulator of ABA signal transduction in a higher plant, which in addition requires three components (receptors, PP2C phosphatases and SnRK2 protein kinases). We have overcome redundancy in the *Arabidopsis* genome using an experimental platform/resource and have characterized and identified this signaling mechanism.

2) We directly answer a long-term open question: "How can SnRK2 kinases be re-activated after de-activation by PP2Cs?" This question has remained open with two possible models that either auto-phosphorylation or phosphorylation by an unknown kinase may reactivate SnRK2 kinases in ABA signaling. Our results clearly show that the identified M3Ks are a missing mechanism and we further show for the first time that auto-phosphorylation cannot alone reactivate the SnRK2 kinases. There is no similar study focusing on this issue to our knowledge. In this regard, this study is very important and adds significantly to our understanding of early ABA signaling, which will be of broad interest to the wider community.

3) ABA receptor PYR/RCAR, PP2C phosphatase and SnRK2 kinase are the minimal core components required for early ABA signal transduction as was found in 2009. Our manuscript adds a new component to early ABA signaling. Our study demonstrates that these three

components are surprisingly insufficient to achieve ABA signal transduction *in vitro* with dephosphorylated SnRK2/OST1. We show using three independent reconstitution systems (Fig 4a-f) that adding M3K δ s enables ABA responses. We have mapped an essential phosphorylation site in OST1/SnRK2.6 that cannot be auto-phosphorylated. This finding adds to the fundamental understanding of early ABA signaling.

4) It has been known since 2006 that early drought/osmotic stress can rapidly activate SnRK2 protein kinases, long before ABA biosynthesis has commenced. However, the molecular mechanisms of the early drought/osmotic sensing machinery remain unknown in plants. Interestingly and unexpectedly, we further have found that the identified M3Ks are essential for the rapid osmotic stress activation of SnRK2 protein kinases. Our findings further point to a new hypothesis that a potential rapid osmotic stress-dependent post-translational modification of M3Ks, provides a possible convergence mechanism for ABA- and osmotic-stress signal transduction. The advances reported in our manuscript will add the newly identified M3Ks to the early ABA and osmotic stress signaling module.

We agree that our present findings point to many new questions that we and the community will investigate. The present study represents over 6 years of research on these M3Ks.

2. In Fig.1d, SnRK2 activity is not completely dismissed in *ami_m3k*. The same with Fig. 6a-d. What does this mean? Some other M3Ks or kinases could be involved in the regulation of SnRK2?

Response:

It is true that the SnRK2 activity in the *ami_m3k* and the *m3k* triple mutants is not completely impaired. We discuss that there may be other M3K(s) that could function redundantly in the *Arabidopsis* genome. There are 22 genes in the *Arabidopsis* M3K B subgroup, some of which may contribute to this pathway. We have now investigated one of the closest homologs (*M3K δ 5*), which is also targeted by both our *amiR-ax1117* and *m3k* amiRNAs isolated in forward genetic screens, and we created a higher-order quadruple mutant. Interestingly, new data show that this mutant shows a stronger ABA-insensitive phenotype (newly added Supplementary figure 16) than the triple mutant as shown below. This result strengthens our findings, however, to analyze this phenotype further and isolate higher order mutants will take much longer and we feel is beyond the scope of this study. In depth analyses of quadruple, quintuple and higher order mutants are important goals for future research.

3. In Fig. 2 and 3, the authors used a mutant form of SnRK2 (S171A) for kinase assay or complementation of *ost1*. However, the presented data is not enough to confirm the biological significance of S171 phosphorylation in ABA signaling. It is highly possible that SnRK2 S171A lost its enzymatic activity, because S171 is located on the kinase activation loop. If so, it is not surprising that SnRK2 S171A did not complement with *ost1* phenotype. Authors should check that some other mutations in kinase activation loop can complement *ost1*.

Response:

Thank you for this suggestion. To answer this question, we created other mutations in OST1/SnRK2.6 activation loop (S175A and T176A) and tested whether they impair kinase activity *in vitro* and ABA activation *in vivo*. As shown below left panel (newly added Supplementary Fig. 5a), *in vitro* phosphorylation assays show that recombinant GST-OST1/SnRK2.6 carrying S171A or T176A mutations have largely intact kinase activity. However, only S171A shows no ABA activation in mesophyll cell protoplasts (right panel; newly added Supplementary Fig. 5b). A different S175A mutation impairs kinase activity both *in vitro* and *in vivo*. These results strengthen our conclusion that OST1-Ser-171 is essential for *in vivo* activation of OST1/SnRK2.6 without disrupting kinase activity itself. We have investigated these activities in transformed mesophyll protoplasts. Creating stable homozygous transgenic lines will take a significant time, and we feel this is beyond the scope of our present manuscript.

4. The authors concluded that M3K phosphorylates Ser171 on SnRK2 in response to ABA. However, there is no evidence of *in vivo* S171 phosphorylation by M3K. It is necessary to check the phosphorylation status of Ser171 in the *m3k* disruptants, for example, MS analysis of immunoprecipitated SnRK2 from WT and *m3k* disruptants, or western blot analysis using an anti-pS171 antibody.

Response:

Thank you for this comment. To address this question, we used the mesophyll cell protoplast (MCP) expression system as we do not have specific OST1/SnRK2 antibodies and generating homozygous transgenic plants will take a long time. OST1/SnRK2.6-GFP was transiently expressed in MCPs and purified by immunoprecipitation using GFP antibodies. We find clear Ser-171 phosphorylation in response to ABA in wild type MCPs by tandem MS/MS analyses (Supplementary figure 6). We also tried to compare phosphorylation levels between wild type and *m3kδ1/δ6-1/δ7* triple mutant; however, our results thus far are noisy and unclear, which may be due to the need to knockout additional M3Ks, as explained above. We believe we will need higher order *m3k* mutants to robustly investigate a lower phosphorylation level in an *m3k* mutant background. This analysis is not easy to conduct, and further analysis will have to await higher order mutants with stronger phenotypes, bulking of such lines, identifying appropriate mutant combinations and in depth analyses. Also, because we have neither stable OST1-overexpressing

lines in the *m3kδ1/δ6-1/δ7* background nor a pS171 antibody these proposed experiments would take a substantial amount of time. To address the reviewer's comment we have added the new data, showing a clear ABA-induced phosphorylation of S-171 (Supplementary figure 6).

5. There is no evidence to show *in vivo* interaction between SnRK2 and M3K.

Response:

We have now pursued experiments to probe *in vivo* binding by co-immunoprecipitation using a transient expression system in protoplasts. However, we did not detect co-immunoprecipitation of M3Kδ6-3xFLAG with OST1-GFP as shown below (Supplementary Fig. 15a). However, we think this does not contradict an *in vivo* interaction between these proteins. In general, kinase-substrate interactions can be dynamic and often produce only transient interactions, which make it difficult to detect their interaction. In some cases, special methods are required. We have also pursued quantitative BiFC experiments, as this method allows detection of transient interactions. We have included known positive BiFC interaction controls and quantified fluorescence intensities. The results show that M3Kδ6 and δ7 interact with some SnRK2 kinases, and quantitative analyses are included. We included these results to our revised manuscript (Supplementary Fig. 15).

[redacted]

6. Phenotyping of *m3k* disruptants in Fig. 5. Why don't you check some other ABA responses in the mutant? For example, stomatal conductance, root elongation, or gene expression. Especially, authors demonstrated that M3Ks are involved in the regulation of SLAC1 activity in Fig. 4, suggesting that M3Ks could have some roles in guard cells. This point needs to be confirmed in planta using M3K disruptants. In summary, it is still unclear whether M3Ks are global regulators of ABA signaling in higher plants or not.

Response:

To address this comment, we carried out phenotypic analyses including ABA-induced stomatal closure. Using a robust intact leaf gas exchange method we recently reported (Ceciliato et al. Plant Methods 2019), we found that *m3k_ami* plants show a strong ABA insensitivity (Supplementary Fig. 8). In addition, we have performed root elongation assays. In this assay, the *m3k δ 1/ δ 6-1/ δ 7* mutant shows a reduced ABA sensitivity in inhibition of primary root elongation on 1/2 MS plates supplemented with ABA. These results strengthen our conclusion that the M3K δ 1, δ 6 and δ 7 function in higher plant ABA responses. We included this result in Supplementary Fig. 11c in the revised manuscript. As noted above, we are isolating higher order *m3k* mutants and bulking these seeds up and we plan to include more in depth analyses in a future study, that is beyond the scope of the present study.

Reviewer #3 (Remarks to the Author):

This submission reports identifying the upstream re-activators of the core SnRK2 kinases in ABA signaling. These activators were revealed as mutations/mutants in the M3K δ s (MAPKK kinases) family generated by ami-RNA that showed reduced ABA sensitivity on seed germination. Allelic mutations in some of these M3K δ 6/7 also showed little or no detectable “SnRK2” activity upon ABA or osmotic treatments.

The authors have provided several lines of evidence to support their hypothesis, using OST1/SnRK2.6 as the representative SnRK2. The SnRK2 kinase OST1/SnRK2.6 expressed in *E. coli* is active (unlike other members), presumably by “autophosphorylation”. But once OST1/SnRK2.6 is inactivated by protein phosphatases (e.g. by ‘non-specific’ CIP or specific HAB), the SnRK2 cannot resurrect itself by auto-phosphorylation. Then, how does OST1/SnRK2.6 gets re-activated? The answer is that this needs upstream kinases, namely the M3K δ . The supporting lines of evidence are the following:

(A) In-gel kinase assays, the protein extracts from the M3K δ mutants (extracts pooled from mutants? I suppose) have reduced activities for at least three kinases, which the authors attributed to SnRK2s.

(B) OST1/SnRK2.6 treated by Calf Intestinal Phosphatase was unable to autoactivate a second time, but needs trans-phosphorylation by M3Ks $\delta 1/ \delta 6/ \delta 7$ on S171, S175 and Thr176 (Fig. 2C). The analyses then concentrated on the S171 site (based on the authors' evidence that it is not autophosphorylated, Fig. 2d).

(C) Functional complementation of the *ost3-1* mutant and ABA-activation of S-type anion channel currents by patch-clamp using guard cell protoplasts from these stably transgenic lines support the importance of S171 for OST1/SnRK2.6 kinase activity (Fig. 2e; Fig. 3).

(D) In vitro reconstitution of the ABA pathway showing that the addition of M3K $\delta 6$ restored the OST1/SnRK2.6 kinase activity (using phosphorylation of a transcription factor as the readout), previously deactivated by the HAB1 protein phosphatase 2C. This is supplemented by reconstitution of the pathway in *Xenopus* oocytes, with the activation of anion channel SLAC1, as the readout.

(E) Disruptions in double or triple mutants of M3Ks $\delta 1/ \delta 6/ \delta 7$ lead to ABA-insensitive seed germination, as compared to the wild type, based on cotyledon greening.

The results are very enlightening and add to previous pharmacological studies that this family of SnRK2s are activated by different upstream mechanisms. One of these is thus via the mitogen-activated kinase pathways that are activated by ABA-dependent and -independent signals.

May I make a couple of remarks that might improve the weight of the evidence and the clarity of the paper?

Major point:

1. Figure 3 showed the functional importance of S171 in the OST1 protein by complementation of the *Arabidopsis* *ost1-3* mutant phenotype (transpiration, S-type anion currents) using either the WT transgene or the counterpart with the S171A mutation. This is fine as a prove of principle. But to connect the functional significance of S171 better with the upstream M3K δ s, and in planta, have the authors attempted to by-pass the M3K δ mutations with the transgene OST1/SnRK2.6 bearing a phospho-mimic Ser171?

Response:

This is an interesting suggestion. To answer this comment, we have now first tested if a "phospho-mimic" mutation causes a constitutively active OST1/SnRK2.6 in plant cells. We prepared a OST1(S171E)-GFP expression vector and transiently expressed it in *Arabidopsis* mesophyll cell protoplasts. However, unfortunately, OST1(S171E) has no detectable kinase activity as shown right (newly added Supplementary Fig. 5c). Phospho-mimic mutations do not always reproduce phosphorylation effects on proteins. These results suggest that other strategies than phosphomimetic mutations would be needed to further characterize S171 in planta. We have pursued tandem mass spectrometry

experiments and have added new evidence showing that ABA causes OST1/SnRK2.6 phosphorylation in wildtype (Supplementary Fig. 6).

2. The phenotypic characterization for the M3K δ mutants could be more complete. Because the *ost1-3* mutant showed an excessive transpiration phenotype, one would expect that the upstream M3K δ to have comparable stomatal dysfunction, assuming that OST1/SnRK2.6 and M3K δ function in the same pathway in planta. If these M3K δ have several SnRK2 targets, have the authors actually observed that the stomatal phenotype in M3K δ triple mutant being even more severe than that in *ost1-3*. (Fujii et al., 2011). More data on stomatal phenotypes and patch-clamp analysis on the M3K δ mutant guard cells should add more solidity to the in planta results.

Response:

We have conducted stomatal conductance assays and found that an *m3k_ami* line shows a clearly reduced ABA sensitivity in stomatal closing in intact plant leaves. We included these results in Supplementary Fig. 8 in the revised manuscript. For patch clamp analysis, in depth patch clamp experiments will take a long time. We are isolating higher order *m3k* mutants that include highly-expressed M3Ks in guard cells, which can take longer time. These experiments also will require experiments conducted by an expert, who has other ongoing projects. We feel this is beyond the scope of this study and are planning future in depth analyses.

Minor queries:

3. I find that the description on mutant isolation not easy to follow. It is not intuitive to relate the 14 mutants back to the three lines that were included in the prior collection of 25 mutants. Are they independent or some/all are from the same parental lines etc? For this reviewer, because a mutational event in a specific locus is mathematically rare in an unbiased screen, many progeny from the same parental lines could like be from the same mutant (for a given phenotype), unless proven otherwise.

Response:

Thank you for pointing out this issue. We found that our sentences in the original manuscript were not sufficiently clear. We improved our explanation as follows: "By unbiased forward genetic screening of seeds from over 1,500 independent T2 artificial microRNA (amiRNA)-expressing lines in pools (\approx 45,000 seeds screened) for ABA-insensitive seed germination, we isolated approximately \sim 290 putative mutant plants. In secondary screening of the surviving putative mutants in the next (T3) generation, progeny from 25 of the putative mutant plants continued to show a clearly reduced ABA sensitivity, including seeds propagated from three *amiR-ax1117-*

expressing plants (Fig. 1a-c). It is most likely that the three *amiRNA-ax1117*-expressing plants were the progeny of the same *amiRNA*-expressing parent line." (Page 2, line 59).

4. Fig. 1d, *ami_m3K*, for the *in-gel* kinase assays. Are the phosphorylation activities from pooled mutant seedlings of the genotypes M3K δ 5, M3K δ 7, M3K δ 1, M3K δ 6, M3K δ -CTR1? Or from a multiple mutant?

Response:

Fig. 1d shows data using the *m3k_amiRNA* mutant isolated several independent times in forward genetic *amiRNA* screening. This *amiRNA* is predicted to target seven *M3K* genes as shown in Supplementary Fig. 1. In the manuscript, we state "We investigated ABA-activation of SnRK2 protein kinase activity in seedlings of the *m3k_amiRNA* line by *in-gel* kinase assays. Interestingly, ABA-activation of kinase activities was reduced by 60% (Fig. 1d, e; n = 3 experiments)" (Page 2, line 76).

5. From the above *in-gel* kinase assays alone, besides the mol. weights of the kinase activities as clues, a non-expert would unlikely jump to the conclusion that the reduced histone phosphorylation signals must correspond to the SnRK2.2, 2.3 and 2.6. Can the authors explain this better? I think histones are also used as a surrogate substrate for many kinases.

Response:

Thank you for this suggestion. Previous studies in several laboratories, including in our lab have found that these kinase activities vanish in *ost1* and *snrk2.2/2.3* mutants. We have added an explanation to the Results section: "SnRK2 protein kinases are known to be detected at apparent mobilities of 40 to 44 kDa in *in-gel* kinase assays (Fujii et al. PNAS 2009; Umezawa et al., PNAS 2009)." (Page 2, Line 77).

6. Fig. 6 comes back to the mutant M3K δ characterization by osmotic stress and ABA. These results maybe better if presented earlier along with Fig. 1d. If the figure is too dense as a consequence, some of the results could be transferred to the supplementary data section.

Response:

As explained above, in Fig. 1d, we used our *m3k_amiRNA* line for *in-gel* kinase assays. Based on this finding, we found M3K δ 1, M3K δ 6 and M3K δ 7 as important contributors through biochemical and electrophysiological analyses (Fig. 1-4), and we created the *m3k* triple mutants (Fig. 5). Then, these results and SnRK2.4 data led to the interesting finding that rapid osmotic stress-induced SnRK2 kinase activation is surprisingly largely impaired in the triple mutants (Fig. 6). This also coincides with the sequence of discovery. If we move Fig. 6 to Fig. 1, this would cause a non-clarity in the logic of the research. For this reason, we still think that Fig. 6 should be coming last in this manuscript. We hope this explanation is acceptable.

7. Besides the one (and older) reference 19 cited in this manuscript, there are two other recent references on SnRK2.4, being targets of the ABA-related PP2Cs (Krzywinska et al., 2016b;

Krzywinska et al., 2016a). There is also a paper by Danquah et al. (2015) that reported an ABA-activated MAP kinase cascade. The authors may want to check them out to see whether they may be worth citing in the same vein of ABA-dependent and ABA-independent signal cross-talk.

Response:

Thank you for the suggestion. We now cite these papers (Page 6, line 251; Page 7, line 280).

8. In Fig. 5d, Fig. 6a, b. Some mutants are designated only by *crispr*; is this the most severe multiple mutant or a pool of mutants created by *crispr*? Please clarify how cotyledon “greening” assays shown in the panels (e) (h) and (j) were quantified. I suppose that this is purely visual, to give a rough indications of the mutant phenotypes on germination. Do the authors think that it might be useful to have quantification of chlorophyll for a given number of germinated plants etc to give some precisions? It appears that some mutant lines are greener than others, and the sizes of the different mutants are different. But this kind of cross comparisons are difficult because these lines were germinated on different concentrations of exogenous ABA, for different lengths of time. Do these M3K mutants display altered seed dormancy (reduced?) as a quantitative indicator of sensitivity to endogenous ABA?

Response:

We targeted and isolated a specific *crispr* triple mutant allele and bulked homozygous seeds of this allele for these experiments. We conducted cotyledon emergence assays by counting visible emerged green cotyledons as an indicator of ABA sensitivity. Because this analysis method is widely-accepted in this field, we had not explained the method in detail. Now we have added a brief explanation in the revision (Page 5, line 189). Using this method, we repeated 3- to 4-independent experiments for statistical analyses. We have also revised wording "Cotyledon greening (%)" to "Green cotyledons (%)" in Figure 5e, h and j to avoid confusion.

[redacted]

References:

Danquah, A., de Zelicourt, A., Boudsocq, M., Neubauer, J., Frei Dit Frey, N., Leonhardt, N., Pateyron, S., Gwinner, F., Tamby, J.P., Ortiz-Masia, D., Marcote, M.J., Hirt, H., and Colcombet, J. (2015). Identification and characterization of an ABA-activated MAP kinase cascade in *Arabidopsis thaliana*. *Plant J.* 82, 232-244.

Fujii, H., Verslues, P.E., and Zhu, J.-K. (2011). Arabidopsisdecuple mutant reveals the importance of SnRK2 kinases in osmotic stress responsesin vivo. *Proc Natl Acad Sci U S A* 108, 1717-1722.

Krzywinska, E., Kulik, A., Bucholc, M., Fernandez, M.A., Rodriguez, P.L., and Dobrowolska, G. (2016a). Protein phosphatase type 2C PP2CA together with ABI1 inhibits SnRK2.4 activity and regulates plant responses to salinity. *Plant Signaling & Behavior* 11, e1253647.

Krzywinska, E., Bucholc, M., Kulik, A., Ciesielski, A., Lichoicka, M., Debski, J., Ludwikow, A., Dadlez, M., Rodriguez, P.L., and Dobrowolska, G. (2016b). Phosphatase ABI1 and okadaic acid-sensitive phosphoprotein phosphatases inhibit salt stress-activated SnRK2.4 kinase. *BMC Plant Biol.* 16, 136.

Reviewer #4 (Remarks to the Author):

Takahashi et al's study firmly places MAP3Ks as a core ABA/osmotic signalling component in *Arabidopsis thaliana*. ABA signalling has been shown by various groups to involve ABA binding to the PYR/PYL/RCAR which inhibits a PP2C phosphatase releasing a SnRK2 kinase to interact/phosphorylate downstream targets. Here the authors find a new component in this pathway. The authors conclude that 3 MAP3Kdelta1/6/7 are required to fully activate SnRK2 kinases for complete ABA signalling to occur and that these kinases are likely to represent a convergence point between (ABA independent) osmotic responses and ABA signalling. This component has been previously overlooked due to the nature of in vitro studies which spontaneously phosphorylate SnRK2s. To resolve the effects authors had to dephosphorylate the SnRK2 to show that 1) they are not autophosphorylated and 2) that specific MAP3K can phosphorylate SnRK2s.

This is a follow up to the Schroeder lab's pioneering 2013 Plant Cell paper where they introduce a screen that identifies plants containing amiRNA can be used to overcome 'redundancy' and identify molecular components involved in plant function. In that work they find plants with amiRNA targeting several MAP3Ks display ABA-insensitive germination compared to wildtype. Here, they fully validate that preliminary observation and go on to show that some of these MAP3K interact with OST1 (SnRK2.6) and phosphorylate S171 (not previously known for OST1 activation) and this is required for ABA induced stomatal closure via activation of SLAC1. Furthermore, they show evidence that MAP3K are also involved in osmotic induced phosphorylation of SnRK2s.

The study is very high quality and I find little fault with the experimental approaches and results as presented. The finding that MAP3Ks likely act as a convergence point for ABA and osmotic signalling is significant.

Major points to address

1) Shitamichi et al 2013 Plant Biotechnology showed that MAP3Kdelta4 overexpression had the same relief of ABA inhibition of germination as the MAP3Kdelta triple knockouts here, it would be appropriate to cite and discuss. Furthermore, Li et al 2015 PCP also showed that germination of Raf10 and 11 (B2 MAP3Ks incl. MAP3Kdelta4) are insensitive to ABA and overexpression plants are more sensitive to ABA. Again this paper is not cited. There are also several more MAP3K papers published from Japan that show involvement of MAP3Ks in ABA signalling albeit not showing the likely interaction with SnRK2s.

Response:

Thank you for these comments. We have added text in the Discussion section and cited Shitamichi et al 2013 Plant Biotechnology; Lee et al., 2015 PCP; Matsuoka et al., 2015 Plant Mol. Biol.; and

Li et al. 2017, Plant Physiol. in the revised manuscript (Page 6, line 251). The Shitamichi et al. 2013 study suggests that MAP3K δ 4 is a negative regulator of ABA signaling which differs from the MAP3Ks we have studied here.

2) This manuscript contains three distinct but related elements to fully demonstrate the involvement of MAP3K in ABA/osmotic signalling. Firstly, the examination of OST1 regulation of SLAC1-facilitated stomatal closure, then the ABA inhibition of germination, and finally osmotic activation of SnRK2s. All point to the involvement of MAP3K. These findings highlight the importance of MAP3K but none of these aspects of the study are fully mature. For instance:

A) what is MAP3K interacting with in roots during the germination responses to ABA – do they involve SnRK2s?

Response:

We have now performed root elongation assays. In this assay, the *m3k δ 1/ δ 6-1/ δ 7* mutant shows a reduced ABA sensitivity in inhibition of primary root elongation on 1/2 MS plates supplemented with ABA. These results strengthen our conclusion that the M3K δ 1, δ 6 and δ 7 function in higher plant ABA responses. These data have been added to the in revised manuscript (Supplementary figure 11c). In roots, SnRK2.2 and SnRK2.3 have an important role in the ABA response. Interestingly, serine residues corresponding to the OST1 Ser-171 are conserved in SnRK2.2 and SnRK2.3. We newly created S -> A mutated SnRK2.2/2.3 at these serine residues, and found that these mutations also impair ABA activation of SnRK2.2/2.3 in plant cells (Supplementary Fig. 14b). Also, we investigated binding between M3Ks and SnRK2s in plant cells using BiFC assays, and the results show that M3K δ 6 and δ 7 interact with OST1/SnRK2.6 and 2.2, 2.4 and 2.10 with different apparent affinities in this system. We included these new results in our revised manuscript (Supplementary Fig. 15b).

B) What are the ABA-related stomatal phenotypes of the MAP3K mutants? i.e are the MAP3K essential to ABA-activation of SLAC1 (as carried out in vitro).

Response:

We have performed stomatal conductance assays using the recently reported method in intact leaves (Ceciliato et al. 2019 Plant Methods). We found that *m3k_ami* plants show a strong ABA insensitivity (Supplementary Fig. 8). We are pursuing isolation of higher order mutants including high guard cell-expressed M3Ks. This can take several years to identify and characterize mutant combinations in depth.

C) What are the plant osmotic related phenotypes in the map3kdelta triple knockouts?

Response:

We showed osmotic stress-insensitive seed germination phenotypes of *m3k* mutants (Supplementary Fig. 6 in the original manuscript). [redacted]

Whilst it cannot be debated that MAP3K are clearly identified as a component in ABA signalling, the firmness of conclusions regarding the exact mechanisms of how they are involved needs several dotted lines of evidence joining. This is especially true as this study convincingly shows that *in vitro* oocyte studies do not to fully represent what occurs in *planta*. Therefore, can the authors fully recapitulate the findings of the *in vitro* experiments carried out here in *planta*? I would have thought this to be very important to support the conclusions? Alternatives (not required – but suggestions) could also be is *ost1* redundant in *map3kdelta* triple knockout lines? or can the *map3kdelta* triple knockout phenotypes be rescued by an activated OST1? Or potentially are there confirmatory phenotypes of overexpression plants?

Response:

Thank you for this comment and suggestions. To answer this comment, we have now first tested if a "phospho-mimic" mutation causes a constitutive active OST1/SnRK2.6 in plant cells. We prepared a OST1(S171E)-GFP expression vector and transiently expressed it in *Arabidopsis* mesophyll cell protoplasts. However, unfortunately, OST1(S171E) has no detectable kinase activity (newly added Supplementary Fig. 5c). Phospho-mimic mutations do not always reproduce phosphorylation effects on proteins. These results suggest that other strategies than phosphomimetic mutations would be needed to further characterize S171 *in planta*. [redacted] These circumstances have slowed this aspect of this research, and it will take a time to complete these experiments. In this manuscript, we used amiRNA-, T-DNA- and CRISPR/Cas9-based mutants along with biochemical analyses *in planta* and *in vitro*. We also show the relevance of the identified M3Ks in three different reconstitution assays and show ABA response phenotypes in mesophyll cell protoplasts, in seed germination, now also in ABA-regulated root growth and stomatal movements. We think these results are strongly supporting our conclusions. We have also analyzed ABA-dependent phosphorylation of OST1/SnRK2.6 (see comment 3 below). To further investigate M3K functions *in planta*, higher order mutants will be needed, for which it can take a long time to investigate different combinations.

3) On that note. In vitro phosphorylation studies may force phosphorylation to occur when it doesn't in the cell. Can the authors show co-expression data/in vivo interaction data as further evidence of the interaction, or do they think the genetic evidence is currently sufficient?

Response:

We have now pursued experiments to probe *in vivo* binding by co-immunoprecipitation using a transient expression system in protoplasts. However, we did not detect co-immunoprecipitation of M3K δ 6-3xFLAG with OST1-GFP as shown below (Supplementary Fig. 15a). However, we think this does not contradict *in vivo* interaction between these proteins. In general, kinase-substrate interactions can be dynamic and often produce only transient interactions, which make it difficult to detect their interaction. In some cases, special methods are required. We have also pursued quantitative BiFC experiments which can detect transient interactions in plants. We have included known positive BiFC interaction controls and quantified fluorescence intensities. The results show that M3K δ 6 and δ 7 interact with some SnRK2 kinases, and quantified fluorescence intensities. We included these results in our revised manuscript (Supplementary Fig. 15).

To further address this question, we have now used mesophyll cell protoplasts (MCPs). OST1/SnRK2.6-GFP was transiently expressed in MCPs and purified by immunoprecipitation using GFP antibodies. We find that ABA clearly causes Ser-171 phosphorylation in wild type MCPs by tandem MS/MS analyses (Supplementary Fig. 6).

[redacted]

As proposed by the reviewer, co-expression data can for example be obtained from eFP browser datasets, with OST1 and three M3Ks copied here:

4) What are the phenotypes of the single and double map3kdelta T-DNA knockouts?

Response:

To answer this question, we performed seed germination assays using single and double mutants. Single mutants only showed potential minor effects. Double mutants show various ABA-insensitive phenotypes as shown below. We have added these data in Supplementary Figure 11a and b.

Other points for consideration

In Fig 1 g, “M3Kdelta1” was labelled twice in the upper panel; Statistical analysis is missing in Fig 1e, 4f, 6b and d.

Response:

We included data showing more than one lane in which M3Kδ1 was added for comparison and each lane is separately labelled.

Regarding statistical analyses, we added statistical analysis data in Figure 4f. However, for *in-gel* kinase assay results in Figure panels 1e, 6b and d, statistical analyses are not typically included in publications because band intensities have to be normalized for each result. For this reason, we did not include statistical analysis in these figures. We have loaded gel lanes side-by-side as controls. This is widely acceptable for these assays. We have however now added an example of repeat *in-gel* kinase experiments in Supplementary figures 2 and 12.

Only M3Kdelta6's data is often been presented not M3Kdelta1 and M3Kdelta7 in several biochemical analysis (such as Fig2 a-b, Fig4 a-c). It's fascinating to see M3Kdelta6 can directly phosphorylate inactive OST1 (D140A), does this also apply to the other two M3Ks? It would be ideal to specify whether M3Kdelta1, 6 and 7 share the same regulation mechanisms in phosphorylating SnRK2s. In Fig 6e, M3Kdelta1 activates SnRK2.2 and 2.3, does this suggest M3Ks have distinctive SnRK2 substrate preference, does this have implications for downstream targets and distinctive roles?

Response:

Thank you for this suggestion. We newly performed *in vitro* phosphorylation assays using M3Kδ1, δ6 and δ7, and found that all of them phosphorylate kinase inactive OST1(D140A) with potential different efficiencies. These apparent efficiencies seem to be consistent with the efficiencies of OST1 activation shown in Fig. 1g. We included these data in Supplemental figure 4.

We also performed *in-gel* kinase assays using GST-SnRK2.3 to check possible substrate preferences of M3K δ 1, δ 6 and δ 7 as shown below. We found that SnRK2.3 is activated by these M3Ks in a similar way to OST1/SnRK2.6 (Fig. 1g). We included these data in the revised manuscript (Supplementary figure 14a).

In Fig 4d, upon co-injection with PYL9+ABI1+OST1+SLAC1, M3Kdelta6 was able to slightly activate SLAC1 anion activity without ABA by eliciting greater inward currents compared to PYL9+ABI1+OST1+SLAC1+M3Kdelta1. In contrast, in Supp Fig 5, all three M3Kdelta1, 6 and 7 were able to activate SLAC1 in a similar manner by interacting with OST1. Can the authors explain this data? Do they think a set of negative control, such as co-expression dephosphorylated OST1 with M3Ks +SLAC1 (OST1S175A+ M3Kdelta6+ SLAC1) would further validate that M3Ks can robustly re-activate dephosphorylated OST1? This could clarify which dephosphorylated residue in OST1 can be fully activated by M3Ks, as authors have mentioned that multiple putative phosphorylation sites have been identified in OST1?

Response:

Three M3Ks activate SLAC1 in the presence of OST1 in Supplementary Figure 5 in a similar way, but only M3K δ 6 slightly activated SLAC1 in the presence of OST1, PYL9 and ABI1. This may be because of a leaky inhibition of ABI1 on OST1 in the presence of M3K δ 6 under the imposed conditions. We did not try the OST1S175A mutant which is well-known to show no kinase activity. However, we have now analyzed a dead version of OST1 (D140A) and did not see any activation of SLAC1 (Supplementary Figure 9f and g).

Does the word core need to be in the title? – it doesn't quite scan easily at present

Response:

Thank you. We have removed the word core from the title to avoid any confusion.

REVIEWERS' COMMENTS:

Reviewer #1 (Remarks to the Author):

The authors partially addressed this reviewer's concerns.

Point 2: Since the authors already generated SnRK2.2 and 2.3 S171 non-phosphorylatable mutants in protoplasts, it should have been relatively easy to test ABA responses such as expression of ABA-responsive genes (or a corresponding reporter gene assay) in protoplasts. I think it is quite likely that the effect on ABA-induced gene expression does not phenocopy the triple deletion as predicted in their model. There has been no need to generate transgenic plants to address this question.

Reviewer #2 (Remarks to the Author):

The revised manuscript has been greatly improved as compared to the previous version. Authors added a lot of new data, especially as supplemental figures to answer reviewer's comments. I think many concerns has been already cleared away, but only one point is still remained as follows.

As authors mentioned, they tried to check in vivo phosphorylation of S171 in OST1 in m3k mutant, but they failed to detect it at present. Unfortunately it is still unclear whether ABA-responsive S171 phosphorylation depends on M3Ks or not, and this point is critical for the main body of author's conclusion. Why don't you try to perform in vivo 32P labeling of SnRK2 using a transient expression system in WT and m3k mutants? This experiment does not require any custom antibodies or stable transgenics.

Alternatively, authors can just weaken their conclusion, e.g. further experiments will be required to confirm S171 phosphorylation by M3K.

Reviewer #3 (Remarks to the Author):

The results in this manuscript contribute important knowledge on how the kinase activity of OST1/SnRK2.6 is reset by the upstream MAPKKK M3Ks $\delta 1/ \delta 6/ \delta 7$ (there might likely to others). The rebuttals to my particular queries were fully satisfactory. It is ashamed that the phosho-mimic mutation in OST1/SnRK2.6 did not work. It would have added whole-plant, and possibly additional electrophysiological evidence to reinforce the model.

Reviewer #4 (Remarks to the Author):

Original reviewer 4 – re-review

I have rereviewed the now updated manuscript by Takahashi et al, on M3K activation of SnRK2s, and their roles in ABA and osmotic signalling.

I am mostly satisfied with the additional data and edits made in the manuscript, and to the responses made to the other reviewers comments.

I only have a few comments specific to the new data and one question I had that has not yet been adequately addressed.

L76 There needs to be a simple one liner indicating why a relationship between M3Ks and SnRK2s was investigated.

The new stomatal conductance data for m3k_ami is a nice addition, but it seems slightly out of place in terms of the run of the manuscript, also it is not commented upon further in the manuscript – it is actually nice linking data. Should it not be included either in the first results paragraph (after the germination phenotype) or after Supp Fig 11?

What line is used in Supplementary Figure 8 – is it amiR-ax117 (with 5 MPK3s knocked down) or m3k amiRNA (with 7 MPK3s knocked out)? I presume it is the later based on the response to comment 4 from reviewer 3.

Can the authors show the true stomatal conductance/transpiration data in Supp Fig 8 (rather than just normalised), are the conductance's of the mk3-ami plants equivalent to ost1 complemented with OST1-S171A? I presume it is higher than wildtype plants?

I think the stomatal conductance of m3k_amiRNA is important and should be mentioned in the discussion – ~Line 262? to extend into the likely physiological role beyond in vitro?

What about the triple m3k knockout, are these the M3Ks involved in ABA signalling in the guard cell or are there additional M3Ks (or additional) redundancy in the pathway? Was this experiment tried? It is not an issue for me if this doesn't have the same ABA insensitivity as the higher order mutant in terms of stomatal conductance (in contrast to your osmotic data) as it is important to define whether further M3Ks are involved in the discrete process of ABA signalling in the guard cell.

In extension to this point, my question in regard to cellular expression was not answered i.e. which M3Ks are in the same cell types as SnRK2.2/3 or 6 the guard cell or roots? This data is available and should be made clear in the manuscript. The deficiencies of in vitro assays in defining components of a signalling pathway are a key point from this work and so to close the loop on which m3ks are likely to interact in planta based on expression is important.

The new data in Supp Fig 6 is very nice showing OST1 residue S-171 is phosphorylated by ABA in mesophyll cell protoplasts. I hesitate to ask this, but it would be very nice to show that this isn't the case in at least the m3k_ami, if not the specific m3k triple knockout, to show dependency upon those M3Ks. I suspect this might be an unreasonable request based on the findings and advances already present in this manuscript. Regardless, what is needed is a few lines further elaborating upon the statement in the text alluding to the need for higher order mutants of the M3K to show this kind of result in vivo. It would be more complete to definitively state whether additional M3Ks are needed to show the effects in the guard cells, or whether further work is needed to fully elucidate whether this is the case. This would close the loop in this paper at least, and make it clear what is now known and not known after this work.

In summary, it was a real pleasure to review this well executed work that has revealed a novel component in a critical signalling pathway in plant environmental responses that was already (thought to be) well characterised.

Reviewer #1 (Remarks to the Author):

The authors partially addressed this reviewer's concerns.

Point 2: Since the authors already generated SnRK2.2 and 2.3 S171 non-phosphorylatable mutants in protoplasts, it should have been relatively easy to test ABA responses such as expression of ABA-responsive genes (or a corresponding reporter gene assay) in protoplasts. I think it is quite likely that the effect on ABA-induced gene expression does not phenocopy the triple deletion as predicted in their model. There has been no need to generate transgenic plants to address this question.

Thank you for reviewing our manuscript. It is an interesting issue to further investigate the function of the newly found phosphorylation sites in SnRK2.2/2.3 corresponding to the OST1 Ser171. Because we provided evidence that the "non-phosphorylatable" mutants SnRK2.2 and SnRK2.3 are not activated in mesophyll cells, ABA-dependent gene expression that requires transcription factor phosphorylation by these specific SnRK2-kinases may be impaired as well. It might be possible to test this using a transient system using *snrk2.2/2.3/2.6* triple mutant protoplasts as this reviewer suggests. Because this triple mutant is impaired in growth, these experiments would take longer to try and we feel that this is beyond the scope of this study and we are planning related future experiments that may be more feasible.

Reviewer #2 (Remarks to the Author):

The revised manuscript has been greatly improved as compared to the previous version. Authors added a lot of new data, especially as supplemental figures to answer reviewer's comments. I think many concerns has been already cleared away, but only one point is still remained as follows.

As authors mentioned, they tried to check in vivo phosphorylation of S171 in OST1 in m3k mutant, but they failed to detect it at present. Unfortunately it is still unclear whether ABA-responsive S171 phosphorylation depends on M3Ks or not, and this point is critical for the main body of author's conclusion. Why don't you try to perform in vivo ³²P labeling of SnRK2 using a transient expression system in WT and m3k mutants? This experiment does not require any custom antibodies or stable transgenics.

Alternatively, authors can just weaken their conclusion, e.g. further experiments will be required to confirm S171 phosphorylation by M3K.

Thank you for reviewing our manuscript. As we explained in the previous revision, we think that higher order mutants of *M3Ks* will be needed to obtain a robust result on M3K-dependent Ser-171 phosphorylation based on our MS/MS experiments. We do not think that *in vivo* ³²P labeling would provide specific information on the specific Ser-171 phosphorylation level under these experimental conditions because of multiple phosphorylation sites in OST1/SnRK2.6. We agree with this reviewer's comment that we should revise our statement. We have added a revised sentence in the Discussion section: "Higher order *M3K* mutants and further experiments will be needed to investigate M3K-dependent Ser171 phosphorylation of OST1/SnRK2.6 *in planta*"

Reviewer #3 (Remarks to the Author):

The results in this manuscript contribute important knowledge on how the kinase activity of OST1/SnRK2.6 is reset by the upstream MAPKKK M3Ks $\delta 1/ \delta 6/ \delta 7$ (there might likely to others). The rebuttals to my particular queries were fully satisfactory. It is ashamed that the phospho-mimic mutation in OST1/SnRK2.6 did not work. It would have added whole-plant, and possibly additional electrophysiological evidence to reinforce the model.

Thank you for reviewing our manuscript. In this revision, as requested we have added additional electrophysiological evidence that we have now obtained showing that ABA-induced S-type anion channel activation is largely impaired in *m3k* amiRNA guard cell protoplasts (Supplementary Fig. 10c-h). These new data further support the presented findings.

Reviewer #4 (Remarks to the Author):

Original reviewer 4 – re-review

I have rereviewed the now updated manuscript by Takahashi et al, on M3K activation of SnRK2s, and their roles in ABA and osmotic signalling.

I am mostly satisfied with the additional data and edits made in the manuscript, and to the responses made to the other reviewers comments.

I only have a few comments specific to the new data and one question I had that has not yet been adequately addressed.

L76 There needs to be a simple one liner indicating why a relationship between M3Ks and SnRK2s was investigated.

Thank you for reviewing our manuscript. We revised a sentence to address this comment: "Because SnRK2 protein kinase activation is a key step in ABA signaling, and based on prior findings described further below (Fig. 1f), we investigated ABA-activation of SnRK2 protein kinase activity in seedlings of the *m3k* amiRNA line by *in-gel* kinase assays."

The new stomatal conductance data for *m3k_ami* is a nice addition, but it seems slightly out of place in terms of the run of the manuscript, also it is not commented upon further in the manuscript – it is actually nice linking data. Should it not be included either in the first results paragraph (after the germination phenotype) or after Supp Fig 11?

In this revision, we moved the stomatal conductance data to after *in vitro* reconstitution of SLAC1 activation in oocytes. Furthermore, with these stomatal conductance data, we also show newly obtained patch-clamp analysis results showing that ABA-induced S-type anion channel activation is impaired in *m3k* amiRNA guard cell protoplasts.

What line is used in Supplementary Figure 8 – is it amiR-ax117 (with 5 MPK3s knocked down) or *m3k* amiRNA (with 7 MPK3s knocked out)? I presume it is the later based on the response to comment 4 from reviewer 3.

Thank you for this question. We used the *m3k* amiRNA line predicted to target seven *M3K* genes. We have updated labels in this figure and legend (Supplementary Fig. 10a and b in this revision) for clarity.

Can the authors show the true stomatal conductance/transpiration data in Supp Fig 8 (rather than just normalised), are the conductance's of the *mk3*-ami plants equivalent to *ost1* complemented with OST1-S171A? I presume it is higher than wildtype plants?

We show the absolute stomatal conductance data (Supplementary Fig. 10a). In these experiments, initial stomatal conductances of *m3k* amiRNA were lower than that of control plants. We currently do not know whether this lower conductance is a phenotype of this amiRNA line that is predicted to target 7 M3K genes or derived from the experimental conditions. We will further investigate this question in future research with higher order mutant combinations. We have now added patch clamp data that further investigate ABA-activation of S-type anion channels, which function in the stomatal closing response (Supplementary Fig. 10c-h) in this revision.

I think the stomatal conductance of *m3k*_amiRNA is important and should be mentioned in the discussion – ~Line 262? to extend into the likely physiological role beyond in vitro?

Thank you for this suggestion. We now mention about the *m3k* amiRNA stomatal conductance in the Discussion section.

What about the triple *m3k* knockout, are these the M3Ks involved in ABA signalling in the guard cell or are there additional M3Ks (or additional) redundancy in the pathway? Was this experiment tried? It is not an issue for me if this doesn't have the same ABA insensitivity as the higher order mutant in terms of stomatal conductance (in contrast to your osmotic data) as it is important to define whether further M3Ks are involved in the discrete process of ABA signalling in the guard cell.

We have tried some gas exchange experiments using *m3k* double and triple mutants. However, results were not very clear suggesting that higher order *m3k* mutants would be needed. We discuss possible reasons as described below.

In extension to this point, my question in regard to cellular expression was not answered i.e. which M3Ks are in the same cell types as *SnRK2.2/3* or 6 the guard cell or roots? This data is available and should be made clear in the manuscript. The deficiencies of in vitro assays in defining components of a signalling pathway are a key point from this work and so to close the loop on which *m3k*s are likely to interact in planta based on expression is important.

We have analyzed a dataset from the public microarray database eFP Browser and created a new chart showing expression levels of six subgroup B3 M3K genes and *OST1/SnRK2.6*, *SnRK2.2* and *SnRK2.3* in guard cells and mesophyll cells (Supplementary Fig. 16b). These data indicate that the *M3K δ 5* gene is strongly expressed in guard cells, implying a possible role in stomatal movements. The *m3k* amiRNA line, which shows an ABA-insensitive stomatal closure (Supplementary Fig. 10), also targets *M3K δ 5* (Supplementary Fig. 1). *M3K δ 5* might function redundantly in ABA-induced stomatal closure with other three M3Ks. We now addressed this with the *m3k* amiRNA stomatal conductance data in the Discussion. We agree that published datasets can be used to build hypotheses for future mutant combinations.

The new data in Supp Fig 6 is very nice showing OST1 residue S-171 is phosphorylated by ABA in mesophyll cell protoplasts. I hesitate to ask this, but it would be very nice to show that this isn't the case in at least the *m3k*_ami, if not the specific *m3k* triple knockout, to show dependency upon those M3Ks. I suspect this might be an unreasonable request based on the

findings and advances already present in this manuscript. Regardless, what is needed is a few lines further elaborating upon the statement in the text alluding to the need for higher order mutants of the M3K to show this kind of result *in vivo*. It would be more complete to definitively state whether additional M3Ks are needed to show the effects in the guard cells, or whether further work is needed to fully elucidate whether this is the case. This would close the loop in this paper at least, and make it clear what is now known and not known after this work.

This is a relevant point that we have addressed and now further expand on in the Discussion. As we had explained in response to another reviewer (Reviewer 2) in the previous revision, we believe that higher order mutants of *M3Ks* will be needed to obtain robust data on M3K-dependent Ser-171 phosphorylation. To address this comment, we revised a sentence in the Discussion section: "Higher order *M3K* mutants and further experiments will be needed to investigate M3K-dependent S171 phosphorylation of OST1/SnRK2.6 *in planta*". Also, as mentioned above, we have added text in the Discussion section to explain the possibility that additional *M3K* gene(s) might be involved in ABA regulation of stomatal movements referring to the *m3k* amiRNA stomatal conductance and public gene expression information.

In summary, it was a real pleasure to review this well executed work that has revealed a novel component in a critical signalling pathway in plant environmental responses that was already (thought to be) well characterised.

Thank you for your constructive comments.